# Naturally Occurring Osteoarthritis Features and Treatments: Systematic Review on the Aged Guinea Pig Model

**DOI:** 10.3390/ijms23137309

**Published:** 2022-06-30

**Authors:** Francesca Veronesi, Francesca Salamanna, Lucia Martini, Milena Fini

**Affiliations:** Complex Structure of Surgical Sciences and Technologies, IRCCS Istituto Ortopedico Rizzoli, Via Di Barbiano 1/10, 40136 Bologna, Italy; francesca.veronesi@ior.it (F.V.); lucia.martini@ior.it (L.M.); milena.fini@ior.it (M.F.)

**Keywords:** spontaneous osteoarthritis, in vivo models, treatments, guinea pigs, Dunkin Hartley

## Abstract

To date, several in vivo models have been used to reproduce the onset and monitor the progression of osteoarthritis (OA), and guinea pigs represent a standard model for studying naturally occurring, age-related OA. This systematic review aims to characterize the guinea pig for its employment in in vivo, *naturally occurring OA* studies and for the evaluation of specific disease-modifying agents. The search was performed in PubMed, Scopus, and Web of Knowledge in the last 10 years. Of the 233 records screened, 49 studies were included. Results showed that within a relatively short period of time, this model develops specific OA aspects, including cartilage degeneration, marginal osteophytes formation, and subchondral bone alterations. Disease severity increases with age, beginning at 3 months with mild OA and reaching moderate–severe OA at 18 months. Among the different strains, Dunkin Hartley develops OA at a relatively early age. Thus, disease-modifying agents have mainly been evaluated for this strain. As summarized herein, spontaneous development of OA in guinea pigs represents an excellent model for studying disease pathogenesis and for evaluating therapeutic interventions. In an ongoing effort at standardization, a detailed characterization of specific OA models is necessary, even considering the main purpose of these models, i.e., translatability to human OA.

## 1. Introduction

Osteoarthritis (OA) is a painful and chronic pathology characterized by joint tissue diseases and loss of function, loss of articular cartilage, sclerosis of the subchondral bone (SB), hyperplasia and inflammation of synovium, and alterations in menisci [1]. 

OA is divided into spontaneous or induced by trauma [2,3]. 

Spontaneous OA due to aging represents the most common form of naturally occurring OA in humans [2,3]. Aging increases the risk of naturally occurring OA development not directly, but by changing the joint structure, increasing the vulnerability to injury, altering chondrocyte anabolic and catabolic activities, and reducing their ability to maintain homeostasis in response to stress. In addition, aging increases cell senescence, which arrests the cell cycle due to oxidative stress, compromises the ability of chondrocytes to maintain the homeostasis of cartilage, and increases proinflammatory mediators [4,5]. 

In clinics, X-ray, magnetic resonance imaging (MRI), and clinical scores are the main diagnostic methods employed for OA, supported by preclinical research on quantification of serum or plasma biomarkers, which is important for early-stage diagnosis, assessment of progression, and therapeutic monitoring [6]. 

To better understand the OA pathophysiological process and treatment outcome, animal models are widely used. 

The two categories of OA in in vivo preclinical models are: (1) spontaneous OA, which is in turn divided into naturally occurring (primary OA) and generated by genetic abnormalities; (2) induced OA, through a surgical procedure (secondary OA) or intra-articular injection of chemical products [7,8].

Small and large animal models are used. Small animals (mice, rats, guinea pigs, and rabbits) are employed for the evaluation of the pathogenesis and pathophysiology of OA, since they show more rapid disease progression and are relatively inexpensive and easy to handle. 

Large animals (dogs, goat/sheep, and horses) are mostly used to study the OA process and treatment. Due to their anatomical similarity with humans, these animals confirm the efficacy of a treatment before clinical employment in trials.

As regards small animal models, they are differently engaged to evaluate spontaneous or induced OA. Among them, mice are employed to better understand the genetic bases of OA, rats are used for the evaluation of chondrocyte metabolism due to metabolic abnormalities, and rabbits are used for tissue bioengineering and focal osteochondral lesions studies.

In addition, rabbits, rats, and mice are useful in studies of induced OA models, through anterior cruciate ligament transection (ACLT), meniscal destabilization, collateral ligament transection, and chemical induction models with monosodium iodoacetate (MIA), papain, collagenase, and quinolones.

The spontaneous OA models show a slow evolution, leading to high economic costs, but they have good correlation with the natural progression of human OA. The naturally occurring models are used to evaluate the pathogenesis of OA, while the genetic models are the result of alterations in genes involved in degradation of the cartilage matrix, in chondrocyte differentiation or apoptosis, in subchondral bone metabolism, and in inflammatory molecules. The genetic models accelerate OA development, but they could induce inter-animal variability and are less representative of the condition present in humans in which several different genetic pathways are modified [3,7,8].

Guinea pigs are used as naturally developed models of OA, in which the effect of age, diet, and sex are investigated as stated by the Osteoarthritis Research Society International (OARSI). Guinea pigs are the most useful for the evaluation of human primary OA and of inflammatory biomarkers for their histopathological and OA lesion similarities with humans. In addition, they quickly reach maturity. This animal shows a structure of the knee joint like the human one, developing a bilateral age-dependent OA with similar progression and histopathological features (chondrocyte death, proteoglycan loss, articular cartilage fibrillation, and osteophyte formation) [9,10]. Among the different strains, Dunkin Hartley (DH) guinea pigs develop OA at a relatively early age compared to other outbred strains. For example, strain 13 guinea pigs develop OA later in their lifespan and have better mobility than DH ones. Guinea pigs with the BS2 strain develop OA around 6 months of age, thus lagging behind those with DH [11]. Considering these characteristics of guinea pigs, they may serve as valuable preclinical models for investigating the etiology of OA and for investigating the efficacy of treatments. 

Therefore, the aim of the present review focused on naturally occurring OA, and we systematically characterized guinea pig animal models for their employment in in vivo studies. More precisely, in the first part of the review, we examined studies that characterized spontaneous OA models in guinea pigs, while in the second part, we evaluated studies of specific treatments used in these models. Particular attention is paid to animal age, type of treatments used, the ex vivo analyses employed to characterize OA and knee joint status, and gender-related differences. 

## 2. Results

### 2.1. Studies Selection and Characteristics

The initial literature search retrieved 233 studies: 69 from PubMed, 110 from Web of Science, and 54 from Scopus. After screening the title and abstract, 217 articles were run through the Mendeley Desktop version 1.17.9 (Mendeley Ltd., London, UK) citation manager to eliminate duplicates. The resulting 118 complete articles were reviewed to establish whether the publication met the inclusion criteria, and 48 were considered eligible for the review. From the reference lists of the selected articles, one additional publication was found, for a total of 49 articles included in the review (Figure 1).

### 2.2. Naturally Occurring OA in Guinea Pigs

Twenty-two studies evaluated the pathological signs of spontaneous OA in the knees of guinea pigs during aging. Fifteen of them performed evaluations in DH guinea pigs: 5 studies in female animals (n = 165), 9 studies in male animals (n = 243), and 1 study compared gender-related differences in 66 males versus 66 female animals. Seven studies compared two different strains of guinea pigs: DH versus BS2 or DH versus Strain 13 (Table 1).

Figure 2 shows the knee joints of DH guinea pigs at two different months of age (12 and 21 months), thus demonstrating different OA stages.

In female guinea pigs, OA was evaluated at different experimental times, from 1 to 12 months of age. In vivo and ex vivo evaluations were prevalently performed through MRI, non-decalcified histological sections, immunohistochemistry (IHC), scanning electron microscope (SEM), transmission electron microscope (TEM), immunoenzymatic assays (ELISA test), microtomography (Micro-CT), and densitometry. Starting from 9 months of age, we observed a reduction in cartilage thickness (CT) and staining and an increase in cartilage degeneration, with a reduction in alcian blue staining and a Mankin score rising with the increase in the animal’s age [12,13]. Before 9 months of age, slight alterations were observed. At 3 months of age, superficial cartilage ondulation [14] associated with mild ulceration and focal PG loss and fibrillation [2] was described. At 6 months, fissures and PG loss in the superficial zone were also observed [14] as well as reduced chondrocyte distribution in the superficial zone [13]. At 10 and 12 months of age, higher MMP13 and Caspase 3, presence of cartilage cracks, hypertrophic chondrocytes, higher COLL X, osteophytes and lower aggrecan and PG content were detected in comparison to younger animals [15]. Additionally, increased osteoclast numbers (stained with TRAP) and pSmad 1/5/8 expression in cartilage and serum estradiol during aging, until 12 months, was observed [2,12,14]. As regards SB, increased bone mineral density (BMD), trabecular thickness and separation (Tb.Th and Tb.Sp), and a decreased trabecular number (Tb.N) were detected during aging [2,14].

In male guinea pigs, OA was evaluated from 1 to 15 months of age principally with non-decalcified histological sections, biomechanics, IHC, gene expression, micro-CT, and ELISA. Regarding cartilage, at 2 and 3 months, knee joints were free from OA signs, with mild irregularities, initial loss of chondrocytes, and increase in Major Histocompatibility Complex (MHC) IIX. At 5 months, we observed severe PG loss, at 6 months increased fibrillation, and at 9 months accentuated hypocellularity, with chondrocyte clustering, tidemark duplication, and fissures. At 12 months, the surface was eroded until complete cartilage loss at 15 months [16,17,18,19]. OARSI and Mankin scores worsened with aging, as well as CT, instantaneous modulus, cellularity, matrix integrity, insulin growth factor 1 (IGF1), and receptor activator of nuclear factor kappa-Β ligand (RANKL) [20,21,22,23]. During aging, as regards SB, an increase in Tb.Th and osteoprotegerin (OPG) and a decrease in bone volume/tissue volume (BV/TV), trabecular bone pattern factor (Tb.Pf), and structure model index (SMI) were observed [17,18,20]. The trabecular bone area decreased in severe OA compared to mild OA with no intraosseous thrombi [16,24]. Finally, in serum, highly regulated on activation, normal T cells expressed and secreted (RANTES) increased and C-Telopeptide of Type II Collagen (CTXII) decreased [20].

The only study that compared male and female animals evaluated SB with micro-CT from 1 to 11 months of age, showing that BMD, BV/TV, and Tb.Th increased with aging in both genders, with higher cartilage degeneration in males than in females [25].

In other studies, DH guinea pigs were compared with BS2 guinea pigs. Already at 2 and 3 months of age, BV/TV, BMD, SB plate thickness, Osterix positive cells, Tb.Th, and dynamic changes in the rod-and-plate microstructure were higher in DH than in BS2 strains [26,27]. On the contrary, SB porosity, SMI, Tb.Pf, Tb.S,p and DA were lower in DH compared to BS2 [26]. During aging, both strains demonstrated an increase in Caspase-3 positive cells, in SB plate thickness, and BMD. Even during aging, SB thickness, Tb.Th, Tb.Sp, BMD, OARSI score, fibrillation, apoptotic cells, cartilage degeneration, and PG loss were higher in DH than in BS2 guinea pigs [11,28,29].

In two studies, DH and Strain 13 guinea pigs were compared. DH males and Strain 13 females showed higher cartilage degeneration compared to DH females and Strain 13 males at 12 months of age. Both strains also showed increased OARSI scores and gastrocnemius collagen synthesis during aging, but DH showed higher OARSI scores and lower fiber angles than Strain 13; in DH alone, reduced gastrocnemius and soleus density and type II myofibers were detected during aging [30,31].

### 2.3. Treatments in Naturally Occurring OA in DH Guinea Pigs

In 27 studies, DH guinea pigs were used to evaluate different treatments in the knee joint: (1) intra-articularly (i.a.) or subcutaneously injective treatments; (2) diet; (3) physical stimulation; (4) mini-osmotic pumps implanted into subcutaneous pockets; and (5) physical activity (Table 2).

#### 2.3.1. Injective Treatments

Epigallocatechin 3-gallate (EGCG) was injected into knees once a week for 3 months, inducing an increase in the running endurance and cartilage COLL II, and a reduction in cartilage roughness and ulceration, osteophyte formation, OARSI score, PG and GAG, MMP13 production, and p16^Ink4a^ protein [32].

Injection of platelet rich plasma (PRP), given once a week for 3aweeks lowered cartilage oligomeric matrix protein (COMP) in synovial fluid, OARSI score, and synovitis, and synovial vascularity was detected at 3 and 6 months after treatment [33]. The effect of 1 or 3 weekly injections of PRP for 3 months was also compared, showing that 3 injections reduced synovitis more than 1 injection [34].

Hyaluronic acid (HA) was added with different concentrations of MSC-recruiting chemokine C-C motif ligand 25 (CCL25) (63, 693, or 6993 pg) and administered once a week for 5 months. CCL25 at concentrations of 693 pg and 6993 pg reduced OARSI scores more than HA alone [35]. Five weeks after injection of HA associated with commercially human mesenchymal stem cells (hMSCs), higher chondrocytes numbers, COLL II, Mankin scores, and smoother surface and lower MMP13 in comparison to HA alone or PBS or MSCs alone [36] were observed. In addition, reprogrammed MSCs from synovial fluid (Re-MSCs) and MSCs from the synovial fluid of OA patients (Pa-MSCs) also reduced TNFα gene expression, while only Re-MSCs reduced TNFα and RANTES proteins and OARSI scores more than Pa-MSCs [37].

In two studies, targeting knockdown IL1β vector (TV) and non-targeting control vector (NTV) or Ad vectors contained coding regions for hIRAP (Ad-hIRAP) and Ad vectors containing coding regions for CMV-driven firefly luciferase (Ad-Luc) were administered in the same animals bilaterally, showing a reduction in TNFα, IL8, Interferon gamma (INFγ), and IL1β expression in cartilage and an increase in transforming growth factor beta 1 (TGFβ1) in TV or Ad-hIRAP compared to NTV or Ad-Luc ones [38,39].

The i.a. injections of Rapa after 1 month reduced the OARSI score MMP13, Glycogenin 1, and Tunel, and increased aggrecan and Beclin 1 compared to 3-methyladenine (3-MA) and saline solution [40].

Weekly i.a. injections of parathyroid hormone PTH (1–34) for 3 months increased GAG and reduced OARSI and apoptosis rates [41], while s.c. injections of PTH (1–34) for 5 days/week for 3 or 6 months reduced the roughness, ulceration, osteophytes OARSI score, MMP13, sclerostin (SOST), and RANKL production, and increased COLL II, PTH receptor (PTH1R), osteoprotegerin (OPG), BV/TV, and SMI more than saline solution treatment [42].

S.c. injections of deferoxamine (DFO), given 2 times a day for 5 consecutive days for 30 weeks, reduced the OARSI score, the expression of mTOR, NF-κB p65, PTGS-2, BAD, BAX, BAK, Caspase-9, Caspase-3, Col2A1, ACAN, MMP2, MMP9, and MMP13, and increased the expression of 4-HNE, p-AMPKα, and TIMP2 [43].

Finally, s.c. injections of risedronate 5 times a week for 6, 12, or 24 weeks reduced OS/BS, BFR/BS, MS/BS, Sb.Th, and serum CTX-II [44].

#### 2.3.2. Diet

Several studies have evaluated DH guinea pigs fed with standard diets enriched with rapamycin (rapa) with or without metformin [45], ad libitum regular or calorie-restricted chow with or without high fat diet (HFD) [46], Oleuropein and/or Rutin with or without Curcumin [47], water with a phosphocitrate analogue CM-01 [48], and D glucosamine (D-GlcN) or chondroitin sulfate sodium (CS) [49].

Eudragit encapsulated rapa with or without metformin was administered for 3 months; both rapa and metformin increased OARSI and PG loss and reduced P-RSP6, while rapa alone increased cartilage damage and decreased P7T AMPK [45].

The effect of calorie restriction was evaluated in animals after 3 months. Calorie restriction lowered superficial PG loss and cartilage MCP1, osteophytes, SB sclerosis, OARSI score, and serum C3 more than in animals fed with HFD [46].

After 8 months, Oleuropein, Rutin, and Curcumin slowed down the progression of spontaneous OA, reducing the OARSI score, synovial score, serum Prostaglandin E2 (PGE2) level, Coll2-1NO2 kinetic curve, serum Fibulin 3 fragments (Fib3-1 and Fib3-2), osteophyte, lining and infiltrated cells, Coll2-1 kinetic curve, and aggrecan neoepitopes (ARGS) and increasing cartilage surface integrity and PG content [47].

CM-01 was administered until 6, 12, and 18 months of age, showing a great contribution to reducing meniscal calcification, osteophyte, surface lesions, PG or chondrocyte loss, cartilage degeneration, Mankin score, and cartilage MMP13, and in increasing CT and cartilage bars in SB [48].

Both D-GlcN and CS Na, given for 8, 12, or 18 months, reduced the Mankin score and MMP3 expression and improved cartilage ECM structure and chondrocytes [49].

#### 2.3.3. Physical Stimulation

Five studies focused on physical stimulation to counteract spontaneous OA development in DH guinea pigs. DH guinea pigs were treated with electroacupuncture (EA) [50,51], hyperthermia [52], pulsed electromagnetic fields (PEMFs) at different frequencies [53], and with pure rapa or liposome-encapsulated rapamycin (L-rapa), at two different doses, combined or not with low-intensity pulsed ultrasound (LIPUS) [54].

For 1 month, from 17 to 18 months of age, EA, administered once every other day, increased mechanical withdrawal threshold, reduced cartilage fibrillation, nucleotide-binding and oligomerization domain-like receptor containing protein 3 (NLRP3), caspase-1 and IL1β proteins, serum TNFα and IL1β, and cartilage MMP13 compared to untreated animals. In addition, NLRP3 inflammasome activation was reduced 2 weeks after treatment [50]. In a second study, EA, given 3 times a week for 3 weeks in 12-month-old DH guinea pigs, improved average speed, maximum speed, total distance traveled and stride length, and the expression of cartilage Col2A1, Fgf18, Tgfb1, Timp1, and Sod2 [51].

Hyperthermia, administered at different time points (one session at 18 months of age with evaluations after 24 h or 7 days, or started from 3 months of age once a month until 9 months of age) increased heat shock protein 70 (HSP70) and reduced the Mankin score, Unc-51-like kinase (ULK1), and Beclin 1 [52].

PEMFs at 37 Hz or 75 Hz were given at 21 months for 3 months. At 24 months, both PEMF frequencies decreased cartilage degeneration, Mankin score, cartilage fibrillation, SB thickness, and Tb.N and increased Tb.Th and Tb.Sp compared to animals that were not treated. The 75 Hz of frequency reduced Mankin score and fibrillation and increased CT compared to the 37 Hz frequency [53]. Figure 3 shows the knee joints of DH guinea pigs at 21 months treated or not with PEMF stimulation.

Animals were also stimulated by LIPUS for 20 min/day, while Rapa or L-rapa were i.a. injected 2 times/week for 2 months, from 6 to 8 months. L-rapa, at both doses, combined or not with LIPUS, showed higher GAG and COLL II and lower OARSI score and MMP13 than control (joints treated with PBS), and L-rapa combined with LIPUS further reduced serum C2C [54].

#### 2.3.4. Mini-Osmotic Pumps

A mini-osmotic pump (every day for 3 months) was implanted into back subcutaneous pockets of DH guinea pigs in three studies.

The pump was filled with a small peptide called T140, a complete CXCR4 inhibitor that can block the SDF-1/CXCR4 signaling pathway, or PBS. T140 reduced the serum stromal cell-derived factor 1 (SDF1), Mankin score, cartilage damage, MMP3, MMP9, and MMP13 expression, and increased COLL II and aggrecan compared to PBS or untreated animals [55].

The effect of T140 was compared with those of TN14003 or AMD3100, the other two antagonists of the SDF-1/CXCR4 signaling pathway. TN14003 induced the lowest serum SDF1 and MMP3, MMP9, and MMP13 cartilage expression, and the highest COLL II and aggrecan expression. Among the other two, T140 exhibited better results than AMD 3100 in mitigating OA signs [56].

In comparison to non-treatment or PBS, AMD3100 reduced cartilage damage, Mankin score, synovial fluid SDF1, GAG, pro-MMP1, MMP13, and serum IL1β [57].

#### 2.3.5. Physical Activity

Physical activity was studied in 18 animals, aged 2 months, on a flatbed treadmill at a rate of 20–25 m/min, for 5 days/week for 22 weeks. It increased aggrecan and reduced the depth of cartilage degeneration in comparison to sedentary condition [58].

#### 2.3.6. Risk of Bias

Risks of bias and quality assessment are summarized in Table 2 and Table 3. Risk of bias of animal studies was high for almost all the examined studies. Among the 49 included in vivo studies, most of them (n = 42) did not declare the method of sequence generation, and in the remaining 7 studies, the method of sequence generation was clearly declared. All the studies showed that groups were similar concerning baseline characteristics (i.e., age, weight, sex) and 8 studies showed that allocation was adequately concealed. Nineteen studies reported that animals were housed randomly during the experiment, and 5 reported the blinding of investigators. Fifteen studies reported that the animals were selected randomly for outcome assessment, and 26 reported the blinding of outcome assessors. All the studies included all the animals in the analyses and reported and detailed the primary outcome, and 4 studies outlined other problems that could result in a high risk of bias.

## 3. Discussion

More than 250 million individuals are affected worldwide and nearly 10–15% of people over 60 years of age suffer from OA with pain and disability. Even if OA can affect all joints, the hip and knee are the most commonly affected (~3.8% of the population worldwide) [1,59].

Although there is no real knowledge about the order of events leading up to OA, the SB thickening seems to be one of the first. SB mineral density is higher in OA patients and leads to sclerosis and osteophyte formation, cartilage degeneration and defects, joint space narrowing, and altered structure and biochemical properties of menisci [60]. Subsequently, hypertrophy and apoptosis of chondrocytes occurs, with a resulting degradation in the cartilage matrix, particularly in Type II Collagen (COLL II). Additionally, extracellular matrix (ECM) degrading enzymes and inflammatory cytokines, such as matrix metalloproteinases (MMPs), a disintegrin and metalloproteinase with thrombospondin type 1 motif 5 (ADAMTS5), interleukin-1 (IL-1), necrosis factor alpha (TNF-α), and nuclear factor kappa-light-chain-enhancer of activated B cells (NF-κB) increase [61,62]. NF-κB acts as a transcription factor that induces gene transcription of many inflammatory cytokines. When it translocates into the nucleus, it triggers the consequent transcription of inflammatory mediators and catabolic enzymes, such as interleukin (IL)-6, IL-8, and MMPs. The prolonged inflammation provokes the loss of the growth-arrested state of articular chondrocytes, deregulated gene expression, and consequent cartilage degradation of OA [63,64].

Currently, several animal models can be replaced with advanced and alternative in vitro ones, but in the case of spontaneous OA models this is not yet possible [65]. Following the 3R (reduction, refinement, and replacement) principles, several infiltrative treatments are tested in advanced in vitro models to ensure the application of an effective and safe product and to reduce the use of animals for scientific purposes. A three-dimensional (3D) in vitro environment provides information about cell–cell and cell–extracellular matrix interactions, as well as cell surface receptor expression, growth factor synthesis, and physical and chemical conditions of a pathology. Several experimental studies highlighted the capabilities of in vitro 3D culture systems to recapitulate the characteristic of a pathology [64,66]. However, in vitro experimentation alone is not sufficient to provide information on the regenerative potential of joint tissues in the OA pathology given the complexity of the microenvironment in which the healing process takes place.

Animal models, by definition, do not perfectly replicate human clinical status. However, much of what is currently known about OA comes from research using models set-up in rodents [67,68], which have yielded promising insight into mechanisms and therapeutics [68]. Specifically, spontaneous OA in guinea pigs shows similarities to the human ones because they develop a bilateral age-dependent knee OA with similar progression and histopathological features [9]. In addition, unlike the other animal species more commonly used in OA studies, there is no need to induce OA through surgery, e.g., meniscectomy or anterior cruciate ligament transection (ACLT), as they exhibit OA with increasing age.

Collectively, the evidence gathered in this systematic review suggests that guinea pigs are a valuable model for both discovery and translational research in OA. Compared to strain 13 and BS2, at the same age, DH showed higher SB and cartilage changes and chondrocyte apoptosis, and it develops OA at an earlier age, thus emerging in this review as the most studied strain of OA (42/49 studies). Within a relatively short period of time, this non-transgenic, outbred model develops aspects of OA including cartilage degeneration and hypertrophic chondrocytes, marginal osteophytes, increased osteoclasts, SB alterations, such as increased BMD, Tb.Th, and TbSp, and a decreased Tb.N. Histologic changes are observed beginning at 3 months of age. Disease severity increases with age, and at 18 months moderate to severe OA is observed (with a Mankin score of ≥11). In contrast, 2-month-old animals had no histologic or radiographically detectable alterations. These structural changes also correlate with biochemical markers, as seen in several analyzed studies by the higher COLL X, MMP13, and Caspase 3 expression. Assessment of biomarkers in serum, urine, or SF further contribute to unraveling correlations between structural changes and biochemical biomarkers with disease stage and structural pathology/histopathology.

Despite only one study having evaluated gender differences in guinea pigs, it was detected that spontaneous/age-associated OA is more severe in males than in females [25]. This may reflect greater weight gain in older males in these species, and these results are consistent with our previous systematic review, in which we evaluated all the preclinical studies dealing with gender differences in animal models affected by OA. In 18 in vivo studies, a higher prevalence of OA and greater bone and cartilage architecture changes were found in males compared to females [69].

In all the studies of the present review, the method of choice to evaluate the results was the histology of non-decalcified tissues (cartilage and synovium), embedded in paraffin. This allowed us to perform histomorphometric measurements (i.e., CT), IHC staining to assess the amount of ECM components, MMPs, pro-inflammatory cytokines and apoptotic proteins production, and semi-quantitative scores, of which the most used are OARSI score and modified Mankin score. The OARSI score ranges from 0 (intact cartilage) to 6 (deformation) [70] and modified Mankin score from 0 (normal cartilage) to 18 (cartilage with clefts extending to zones of calcified cartilage, loss of staining, hypocellularity and tidemark reduplicated) [71]. Micro-CT was largely employed for the evaluation of several different SB parameters. Immunoenzymatic (ELISA) assays, performed on serum or SF specimens, were used for the evaluation of OA markers, pro-inflammatory cytokines, cartilage degradation products, and MMP amounts. Gene expression of cartilage structure protein, MMPs, growth factors, pro-inflammatory cytokines, and apoptotic factors were evaluated through RT-PCR. MRI, WB, SEM, or TEM of medial tibial plateau, and biomechanics (instantaneous modulus, indentation test), macroscopic score, and DXA for evaluation of subchondral BMD were also used in some studies.

Because of the similarities to human OA, different pharmacological and non-pharmacological treatment strategies have been evaluated in this model. These strategies range from intra-articular or subcutaneous injective treatments, diet, physical stimulation, mini-osmotic pumps, and physical activity. The most studied treatments were the injective ones, especially those administered intra-articularly. Among them, the orthobiological ones were the most used. PRP were shown to reduce synovitis, MSCs combined with HA increased COLL II and improved cartilage structure, and Re-MSCs demonstrated a reduced TNFα expression. A second group of treatments was represented by diet. In this review, it was detected that a calorie-restricted diet improved cartilage structure and reduced osteophytes and SB sclerosis than an HFD. Several components administered with the diet including rapa, metformin, oleuropein, rutin, curcumin, CM-01, GlcN, and CS were useful in improving cartilage structure and in reducing MMP production. Furthermore, studies on physical stimulation with EA, hyperthermia, PEMFs, and LIPUS were studied. These stimuli act not only on cartilage physical structure, but also on gene expression and protein production. The cartilage structure also improved when mini osmotic pumps, filled with antagonists of the SDFF1/CXCR4 signaling pathway, were implanted into back subcutaneous pockets of guinea pigs. Finally, physical activity in comparison with the sedentary lifestyle, even if evaluated in only one study, showed an improvement in cartilage structure.

Recently, the regenerative capacity of articular cartilage and the restoration of its functions have been identified as achievable goals in OA. The role of MSC residents in the synovium and SF in the regeneration process of articular cartilage was highlighted, and the senescence of chondrocytes has been recognized in the development of OA [72]. Several mechanisms associated with aging are involved in the development of cellular senescence and consequently of OA, including mitochondrial dysfunction, oxidative stress, and replicative senescence [73]. However, the studies, included in the present review, did not evaluate the key markers of cellular senescence and autophagy, the structural proteins whose nuclear expression precedes the secretion of senescence-associated secretory phenotype (SASP) and cellular oxidative stress markers. In addition, no study reported clinical symptoms even in the most advanced ages when OA signs were very severe.

So, future in vivo studies that will evaluate senescence markers and gait analysis will be necessary.

## 4. Conclusions

This review highlighted that (1) DH is the most appropriate strain for the study of spontaneous OA, and age could range from 3 to 18 months, depending on the desired degree of OA. We posit that the OA characterized in DH guinea pigs will provide a valuable platform for contributing to the evolving understanding of OA etiology, including interventions to improve mobility, independence, and quality of life in humans. (2) Histology of non-decalcified tissues remains the best choice to study joint tissues, with IHC staining and semi-quantitative histological scores. (3) Among treatments, the injections of orthobiological are the most studied. (4) Few studies compare male and female animals to evaluate gender-related differences in response to a therapy or in OA development. Thus, other in vivo studies are necessary to elucidate this aspect to improve diagnosis and treatment of OA in the interest of gender-based protocols in an era of personalized medicine.

All these aspects underlined that guinea pigs represent a powerful in vivo model for studying disease pathogenesis and for evaluating therapeutic interventions for OA. However, it is important to underline that there is no single ‘ideal’ model that permits the investigation of all OA characteristics, and consideration of the advantages and disadvantages of each model is to be taken into consideration when designing a study. The continuing identification and development of suitable OA models are still needed, and substantial work remains before the results from this model and other in vivo OA models will be directly translatable to human OA.

## 5. Materials and Methods

### 5.1. Eligibility Criteria

The PICOS (Population, Interventions, Comparisons, Outcomes, Study) model was used to formulate the questions for this study: (1) studies that consider OA guinea pigs (population); (2) studies that evaluate spontaneous naturally occurring OA (interventions); (3) studies that have control interventions (comparisons); (4) studies reporting the effects of spontaneous OA in cartilage and/or bone (outcomes); and (5) preclinical in vivo studies (study design) [74,75].

The question was: ‘What are the main effects and characteristics of spontaneous OA in Guinea pigs?’ Studies from 28 February 2012 to 28 February 2022 were included in this review if they met the PICOS criteria.

As reported in the Appendix A, we excluded the following studies: (1) those investigating OA induced in animals other than guinea pigs; (2) those that studied diseases other than OA or were not focused on cartilage tissue; (3) those that did not study spontaneous OA; (4) in vitro studies; and (5) those that did not study knee OA. Additionally, we excluded case reports, abstracts, editorials, letters, comments to the editor, reviews, meta-analyses, book chapters, and articles not written in English.

### 5.2. Information Sources and Search Strategies

This literature review involved a systematic search conducted on 28 February 2022, according to the Preferred Reporting Items for Systematic Reviews and Meta-Analyses (PRISMA) statement [76]. The search was carried out on 3 databases—PubMed, Scopus, and Web of Science—to identify preclinical in vivo studies of spontaneous OA in guinea pigs. The search was conducted combining the terms “Guinea Pigs” AND “Osteoarthritis” OR “Osteoarthritis, Knee”. In addition, reference lists of relevant studies were searched for other potentially appropriate publications.

### 5.3. Studies Selection and Data Extraction

Possible relevant articles were screened on the basis of the title and abstract by two reviewers (FV and FS), and articles that did not meet the inclusion criteria were excluded. Then, articles were submitted to a public reference manager (Mendeley Desktop version 1.17.9, Mendeley Ltd., London, UK) to eliminate duplicates. Subsequently, the remaining full-text articles were retrieved and examined by the two reviewers (FV and FS), and any disagreement was resolved through discussion until a consensus was reached or with the involvement of a third reviewer (MF).

Data from the retrieved studies were tabulated (Table 1 and Table 2) taking into consideration preclinical in vivo studies that evaluated spontaneous OA in guinea pigs (Table 1) and treatments (Table 2). The results of risk of Bias [77] and Quality assessment [78] of the in vivo studies are summarized in Table 3 and Table 4.

**Table 1 ijms-23-07309-t001:** Preclinical in vivo studies evaluating naturally occurring OA in guinea pigs.

ANIMALS (Age)	AIM	EVALUATIONS	RESULTS	REF.
21 female DH guinea pigs (3, 5 and 9 mo)	To describe the age dependent cartilage degeneration in OA progression	MRI Histology/histomorphometry	At 9 mo ↓ stain intensity, CT; ↑ cartilage degeneration	[12]
12 female DH guinea pigs (6.5, 8, 9.5 and 11 mo)	To describe structural features of OA and the progressive changes during aging	Micro-CT Histology/histomorphometry	Continuous OA changes gradually occur from 9.5 mo. At 6.5 mo ↓ distribution of chondrocytes in superficial zone; At 8 mo rough and irregular surface. At 9.5 mo chondrocytes disappeared in cartilage. At 11 mo depletion of cartilage in the adjacent region with the SB. During ageing ↓ Alcian blue; ↑ Mankin score	[13]
60 female DH guinea pigs (8, 10 and 12 mo)	To observe chondrocyte and matrix degradation in the superficial surface cartilage during aging	Histology/histomorphometry IHC ELISA of serum WB	At 10 mo ↑ surface hyperthrophic chondrocytes, COLLX, MMP13, Caspase3. At 12 mo cartilage not stained in some areas, damaged with longitudinal cracks, hypertrophic chondrocytes, empty cartilage lacunae; ↓ Aggrecan; ↑ COLL X, MMP13, Caspase3, OARSI score, serum CTXII	[15]
30 male DH guinea pigs (2, 3, 5, 9 and 15 mo)	To evaluate joint changes in spontaneous OA	Micro-CT Histology/histomorphometry	At 2 and 3 mo mild articular surface irregularities, slight ↓ PG. At 5 mo severe PG loss, occasional fissures, mild hypocellularity. At 9 mo more accentuated hypocellularity, chondrocyte clustering, tidemark duplication. At 15 mo complete loss of cartilage. During ageing ↑ OARSI score, TbTh; ↓ BV/TV; ↑ Cortical thickness	[17]
40 male DH guinea pigs (1, 3, 6 and 9 mo)	To investigate the association between changes in SB and sGAG content of articular cartilage in spontaneous OA	Micro-CT Histology/histomorphometry Raman Spectroscopy	At 3 mo ↓ chondrocytes. At 6 mo ↓ chondrocytes; ↑ fibrillation. At 9 mo chondrocyte and tidemark disappeared; ↑ fissures. During ageing ↓ sGAG, TBPf, SMI; ↑ BMD, SBT, phosphate, TbTh	[18]
24 male DH guinea pigs (2, 3, 5 and 7 mo)	To evaluate changes in the quadriceps skeletal muscle in spontaneous OA	Histology/histomorphometry ELISA of serum RT-PCR ICDH and LDH enzyme activities	At 2 mo free from OA. At 3 mo ↑ MHC IIX 3, 5 mo: PG loss in mild zone, cartilage surface irregularities; ↑ serum RANTES. 7 mo: ↑ PG loss and irregularities. During ageing: ↓ serum CTXII	[19]
18 male DH guinea pigs (2.5, 5 and 7.5 mo)	To determine structural changes and OPG/RANKL during development of OA	Micro-CT Histology/histomorphometry IHC	During ageing ↑ SBT, OPG, cartilage fibrillation, PG loss, OARSI score; ↓ RANKL, cartilage cellularity	[20]
60 male DH guinea pigs (1, 3, 5, 7 and 9 mo)	To study the progression of spontaneous OA	Histology/histomorphometry Biomechanics	During ageing ↑ OARSI score, instantaneous modulus; ↓ CT	[22]
10 male DH guinea pigs (21.2 ± 2.9 mo)	To characterize association between SB circulation and bone structure and cartilage degeneration in spontaneous OA	Macroscopy Histology/histomorphometry Subchondral vessels IHC	Mild OA than severe OA, no intraosseous thrombi	[24]
66 male and 66 female DH guinea pigs (1–11 mo)	To investigate the structural alterations in spontaneous OA	Micro-CT	From 5 mo ↑ degeneration of the articular cartilage. 1–6 mo ↑ BMD, BV/TV, TbTh. Male ↑ cartilage degeneration than female	[25]
18 DH and 18 BS2 guinea pigs (1, 2 and 3 mo)	To investigate the spatial and temporal SB change of spontaneous OA at early stage	Micro-CT Histology/histomorphometry IHC	DH ↑ SB BMD, SBT, BMD, BV/TV, Osterix positive cells, TbTh; ↓ porosity, SMI, TbPf, TbSp, DA than BS2	[26]
21 female DH and 21 female BS2 guinea pigs (1, 2 and 3 mo)	To evaluate dynamic changes in the rod-and-plate microstructure	Micro-CT Histology/histomorphometry	DH ↑ BV/TV than BS2. DH show dynamic changes in the rod-and-plate microstructure	[27]
24 male DH and 24 male BS2 guinea pigs (10 wks and 4, 6 and 7.5 mo)	To determine the role of chondrocyte apoptosis in spontaneous OA	Histology/histomorphometry IHC	DH ↑ OARSI score, fibrillation, PG loss, apoptotic cells than BS2. During ageing ↑ caspase-3 positive chondrocytes	[28]
24 male DH and 24 male BS2 guinea pigs (10 wks and 4, 6 and 7.5 mo)	To investigate the association between bone remodeling and cartilage degradation with chondrocyte apoptosis in spontaneous OA	Micro-CT Histology/histomorphometry IHC	DH ↑ TbTh, TbN, BMD, BV/TV, cartilage degradation; ↓ TbSp than BS2. During ageing ↑ TbTh, BMD, cartilage degradation, chondrocyte apoptosis	[11]
24 male DH and 24 BS2 guinea pigs (10 wks and 4, 6 and 7.5 mo)	To investigate the temporal and the spatial relationship between bone remodeling in Sbp and Tb in spontaneous OA	Micro-CT	DH ↓ GpTh during aging. DH ↑ SbpTh, TbTh, BMD than BS2. During ageing growth plate completely closed at 7.5 mo; ↑ SbpTh, TbTh	[29]
10 male and female DH and 10 male and female Strain 13 guinea pigs (12 mo)	To determine the association of cartilage degeneration with subchondral BMD in spontaneous OA	Macroscopy Histology/histomorphometry DXA Atomic absorption spectrophometry	DH male and Strain 13 female ↑ severe cartilage degeneration, surface fibrillation, cartilage clefts, PG loss, Mankin score, BMD than DH female and Strain 13 male	[30]
18 male DH and 18 male Strain 13 guinea pigs (5, 9 and 15 mo)	To evaluate skeletal muscle dysfunction, articular cartilage degeneration, and bone loss during aging in spontaneous OA	MRI Fiber angle Histology/histomorphometry IHC Skeletal muscle protein synthesis rates	DH ↑ OARSI, gastrocnemius and soleus mass; ↓ fiber angle than Strain 13. DH ↓ gastrocnemius and soleus density, type II myofibers, gastrocnemius and soleus myofibrillar, cytosolic, and mitochondrial fractional synthesis rates at 15 mo. During ageing ↑ OARSI; ↓ gastrocnemius collagen synthesis	[31]
40 female DH guinea pigs (1, 3, 6, 9 and 12 mo)	To evaluate age-related changes in articular cartilage, BMD, and estradiol levels	ELISA of serum Histology/histmorphometry SEM and TEM IHC DXA	At 3 mo mild ulceration, focal PG loss and fibrillation, formation of microcilia. At 6 mo ulcerations, matrix loss, chondrocyte hyperthrophy and death, rough surface. At 9 mo ostephyte initial development, fibrillation and PG loss, collagenous fibers degenerated into thick bundles and cracks. At 12 mo ↑ ulcerations, matrix loss, ostephytes, erosion, severe cracks. During ageing ↑ Mankin score, MMP3, SB BMD, serum estradiol; ↓ GAG	[2]
32 female DH guinea pigs (3, 6, 9 and 12 mo)	To evaluate TGF-β activity in OA progression	Micro-CT Histology/histomorphometry IHC	At 3 mo superficial zone undulation with matrix loss, pSmad2/3 in all zones. At 6 mo fissures; ↓ PG in superficial zone; ↑ SbPTh. At 9 mo fissures; ↓ PG in middle zone. At 12 mo fractures in deep zones, severe loss of PG, cracks. During ageing: ↑ Mankin score, TbTh, TbSp, TRAP staining, pSmad1/5/8; ↓ TbN, DA, pSmad2/3	[14]
20 male DH guinea pigs (2, 4, 8 and 12 mo)	To evaluate the expression of PPARγ, and H- and L-PGDS during spontaneous OA	Histology/histomorphometry IHC RT-PCR	At 4 mo minor surface irregularity, initial Safranin O decrease. At 8 mo surface erosion, markedly reduced cellularity and Safranin O, initial development of osteophytes. At 12 mo severe surface erosion, PG loss, osteophytes formation, SB sclerosis. During ageing ↓ PPAR γ; ↑ L-PGDS	[16]
16 male DH guinea pigs (6 and 12 mo)	To determine the association between cartilage IGF-1 with loss of chondrocyte and ECM breakdown in spontaneous OA	Histology/histomorphometry RT-PCR	During ageing ↓ cellularity, matrix integrity, IGF-1; ↑ cartilage disruption, PG loss, Mankin score	[21]
25 male DH guinea pigs (1, 3, 6, 9 and 12 mo)	To investigate the expression of MGP in spontaneous OA	Proteomics analysis Histology/histomorphometry IHC	During ageing ↑ Mankin score, MGP than 1, 3, 6, 9 mo	[23]

↑ = increase; ↓= decrease.

**Table 2 ijms-23-07309-t002:** Preclinical in vivo studies evaluating treatments in naturally occurring OA in DH guinea pigs.

ANIMALS (Age)	TREATMENTS	EVALUATIONS	RESULTS	REF.
INJECTIVE TREATMENTS				
24 female DH guinea pigs (3 and 6 mo)	i.a. injections of PBS or EGCG 1/wk for 3 mo	Endurance test Macroscopy Histology/histomorphometry IHC	EGCG ↑ running endurance, COLL II; ↓ roughness, ulceration, osteophyte, OARSI score, surface erosion, PG and GAG loss, MMP13, p16^Ink4a^. During ageing ↓ running endurance, GAG, COLL II; ↑ OA severity with erosion, ulceration, osteophyte, fissures, fibrillation, PG loss, OARSI, MMP13, p16^Ink4a^	[32]
24 male DH guinea pigs (3 and 6 mo)	i.a. injections of saline solution or PRP 1/wk for 3 times	ELISA Histology/histomorphometry	PRP ↓ COMP, OARSI score, synovitis score, synovial vascularity	[33]
32 DH guinea pigs (8 and 11 mo)	1 or 3 (1/wk) i.a. injections of PRP	Histology/histomorphometry	At 8 mo: PRP ↓ synovitis. 3 PRP injections ↓ articular cartilage damage. At 11 mo: 3 PRP injections ↓ synovitis	[34]
42 DH guinea pigs (3 and 11 mo)	i.a. injections of HA ± CCL25 (63, 693 or 6993 pg) 1/wk for 5 mo	Histology/histomorphometry	HA + CCL25 (693 and 6993 pg) ↓ OARSI score than HA alone	[35]
60 male DH guinea pigs (7 mo)	i.a. injections of PBS, HA, PBS + hMSCs or HA + hMSCs	Macroscopy Histology IHC WB	HA weak matrix staining and cell depletion. PBS, HA and PBS + hMSCs ↑ rough surface. PBS, PBS + hMSCs ↓ chondrocytes, matrix fibrillation. HA + hMSCs ↑ numbers of chondrocytes with cluster formations, smooth surface, COLL II, Mankin score; ↓ Macroscopic score, MMP13	[36]
9 DH strain 051 guinea pigs	i.a. injection of Pa-MSCs or Re-MSCs	ELISA Histology/histomorphometry RT-PCR	Re-MSCs ↓ TNFα, RANTES, OARSI score. Pa-MSCs ↓ TNFα	[37]
6 male DH guinea pigs (2 mo)	i.a. injections of TV and PBS or TV and NTV for 2 mo	RT-PCR	TV ↓ IL1β expression than PBS and NTC	[38]
32 male DH guinea pigs (2 mo)	i.a. injections of TV and NTV, Ad-Luc and Ad-hIRAP or TV and PBS for 2 or 4 mo	Histology/histomorphometry IHC RT-PCR	TV and Ad-hIRAP ↓ TNFα, IL8, INFy, IL1β expression; ↑ TGFβ1 than NTV and Ad-Luc. TV ↓ TNFα, IL8, MMP13 expression than PBS	[39]
50 female SPF-grade DH albino guinea pigs (9, 10 and 11 mo)	i.a. injections of rapa, saline or 3-MA	Histology/histomorphometry Macroscopy FMT IHC RT-PCR	3-MA ↑ OARSI score, MMP13, Glycogenin 1, Caspase3, Tunel; ↓ Aggrecan, Beclin 1. rapa ↓ OARSI score, MMP13, Glycogenin 1, Caspase3, Tunel; ↑ Aggrecan, Beclin 1	[40]
27 male DH guinea pigs (6 and 7 mo)	i.a. injections of PTH (1–34) for 3 mo	Micro-CT Histology/histomorphometry IHC	PTH (1–34) ↑ GAG; ↓ OARSI score, apoptosis rate	[41]
48 female DH guinea pigs (1 and 3 mo)	s.c. injections of saline solution or PTH for 3 or 6 mo	Histology/histomorphometry IHC Micro-CT	During aging ↑ MMP13, SOST; ↓ COLL II. PTH ↓ roughness, ulceration, osteophytes, OARSI score, MMP13, SOST, RANKL; ↑ COLL II, PTH1R, OPG, OPG/RANKL ratio, BMD, BV/TV, SMI	[42]
16 male DH guinea pigs (3.5 mo)	s.c injection of sodium lactate solution or DFO for 7.5 mo	CBC Histology/histomorphometry RT-PCR IHC Overhead enclosure monitoring	DFO ↑ OARSI score, mTOR, NF-κB p65, PTGS-2, BAD, BAX, BAK, Caspase-9, Caspase-3, COLL II, ACAN, MMP2, MMP9, MMP13; ↓ 4-HNE, p-AMPKα, TIMP2, change in distance traveled, change in average speed	[43]
48 male DH guinea pigs (1.5 and 3 mo)	s.c. injections of risedronate for 1.5, 3 or 6 mo	Macroscopy Histology/histomorphometry ELISA Indentation test	Risedronate ↓ OS/BS, BFR/BS, Sb.Th, serum CTX-II, MS/BS	[44]
		**DIET**		
27 male DH guinea pigs (5 mo)	Rapa ± metformin for 3 mo	Micro-CT Histology/histomorphometry IHC WB	Rapa ± metformin ↑ OARSI score, PG loss, cortical th; ↓ P-RPS6. Rapa + metformin ↑ cartilage damage; ↓ P/T RPS6. Rapa ↓ P/T AMPK than rapa + metformin	[45]
24 male DH guinea pigs (2 mo)	Calorie-restricted, HFD or calorie-restricted HFD for 3 mo	Micro-CT Histology/histomorphometry IHC ELISA	Regular chow ↑ surface fibrillation, fissures. Calorie-restricted mild superficial PG loss, no MCP1. HFD superficial PG loss, occasional chondrocyte clustering, and focal cell loss. Calorie-restricted ↓ Micro-CT scoring system, small enthesophytes and/or osteophytes, SB sclerosis, OARSI score, serum C3, cartilage MCP1 than HDF	[46]
65 DH guinea pigs (1 mo)	Oleuropein, Rutin or Rutin + curcumin for 8 mo	ELISA Histology/histomorphometry	Oleuropein, Rutin or Rutin + curcumin ↓ OARSI score, synovial score, serum PGE2, Fib3-1 and Fib3-2, Coll2-1NO2 kinetic curve, cellularity; ↑ surface integrity, PG during aging. Oleuropein ↓ osteophyte, lining and infiltrated cells. Rutin or Rutin + curcumin ↓ Coll2-1 kinetic curve, ARGS. Rutin + curcumin ↓ Fib3-1 kinetic curve. During aging ↓ serum ARGS	[47]
21 DH giunea pigs (3, 6 and 12 mo)	CM-01 until 6, 12 or 18 mo	Radiograph Histology/histomorphometry IHC	CM-01 ↓ meniscal calcification, osteophytes, surface lesions, PG and chondrocyte loss, cartilage degeneration, Mankin score, MMP13; ↑ cartilage bars in SB, CT	[48]
50 female DH guinea pigs (3 wks, 1 and 8 mo)	D-GlcN or CS Na until 8, 12 or 18 mo	Histology/histomorphometry RT-PCR	D-GlcN and CS Na ↓ Mankin score, MMP3; ↑ total RNA, chondrocytes. During ageing ↑ TUNEL-positive cells, MMP8; ↓ RNA, ACAN, COLL II, MMP13	[49]
		**PHYSICAL STIMULATION**		
18 male DH guinea pigs (3 and 17 mo)	EA for 1 mo	Nociceptive Behavioral Test Histology/histomorphometry WB ELISA	EA ↑ mechanical withdraw threshold; ↓ fibrillation, NLRP3, Caspase-1 and IL1β, serum TNFα and IL1β, cartilage MMP13	[50]
10 male DH guinea pigs	EA for 3 wks	Open-field enclosure monitoring parameters Treadmill-based gait analysis Histology/histomorphometry ELISA RT-PCR	EA ↑ average speed, maximum speed, total distance traveled, stride length, COLL II, FGF18, TGFβ1, TIMP1, SOD2	[51]
25 female DH guinea pigs (5, 9 and 18 mo)	Hyperthermia	Histology/histomorphometry IHC	Hyperthermia ↑ HSP70; ↓ Mankin score, ULK1, Beclin1 positive cells	[52]
15 DH guinea pigs (21 mo)	PEMFs at 37 or 75 Hz for 3 mo	Histology/histomorphometry	PEMFs at 37 and 75 Hz ↓ cartilage degeneration, Mankin score, FI, SBT, TbN; ↑ TbTh, TbSp. PEMFs at 75 Hz ↓ Mankin score, FI, TbTh; ↑ CT, TbN than PEMFs at 37 Hz	[53]
56 DH guinea pigs (6 mo)	Pure rapa ± LIPUS, L-rapa ± LIPUS, or LIPUS for 2 mo	Histology/histomorphometry IHC ELISA Cell count AST, ALT, BUN, creatinine, electrolytes sodium, potassium, calcium, inorganic phosphorous, chloride	L-rapa ± LIPUS ↑ GAG, cartilage COLL II; ↓ OARSI score, cartilage MMP13. L-rapa + LIPUS ↓ serum C2C. Pure rapa ± LIPUS, L-rapa ± LIPUS, or LIPUS normal cell count, serum AST, ALT, BUN, creatinine and sodium, potassium, calcium, inorganic phosphorous, chloride	[54]
		**MINI-OSMOTIC PUMPS**		
86 male DH guinea pigs (6 mo)	A subcutaneous mini-osmotic pump filled with TN14003, T140 or AMD3100 for 3 mo	ELISA RT-PCR WB	TN14003 ↓ serum SDF1, MMP3, MMP9, MMP13; ↑ COLL II, ACAN. T140 ↓ serum SDF1, MMP3, MMP9, MMP13; ↑ COLL II, ACAN than AMD3100. AMD3100 ↓ serum SDF1, MMP3, MMP9, MMP13; ↑ COLL II, ACAN than no treatment	[55]
36 male DH guinea pigs (9 mo)	A subcutaneous mini-osmotic pump filled with T140 or PBS for 3 mo	ELISA Histology/histomorphometry RT-PCR WB	T140 ↓ serum SDF1, OA changes and severity, Mankin score, cartilage damage, MMP3, MMP9, MMP13 expression, serum SDF1; ↑ COLL II, ACAN expression, COLL II protein	[56]
35 male DH guinea pigs (9 mo)	A subcutaneous mini-osmotic pump filled with AMD3100 or PBS for 3 mo	ELISA DMMB assay Macroscopy Histology/histomorphometry	PBS deep and wide fissures. AMD3100 ↓ cartilage damage, Mankin score, GAG, SDF1, pro-MMP1, active MMP13, IL1β of SF and serum	[57]
		**PHYSICAL ACTIVITY**		
36 male DH guinea pigs (2 mo)	Physical activity on a flatbed treadmill at a rate of 20–25 m/min, 5 days/wk for 22 wks	Histology/histomorphometry Biomechanics Biochemistry	Physical activity ↑ Aggrecan; ↓ depth of cartilage degeneration	[58]

↑ = increase; ↓= decrease; AMPK = AMP-activated protein kinase; ARGS = aggrecan neoepitopes; AST = aspartate aminotransferase; BAD = BCL-2-associated death promoter; BAK = BCL-2 homologous antagonist killer; BAX = BCL-2-associated x protein; BFR/BF = bone formation rate; BMD = bone mineral density; BS2 = Bristol Strain-2s; BUN = blood urea nitrogen; BV/TV = bone volume/tissue volume; C2C = type II collagen cleavage; CBC = Complete Blood Count; CCL25 = MSC-recruiting chemokine C-C motif ligand 25; CM-01 = Carolinas Molecule-01; COLL = collagen; COMP = cartilage oligomeric matrix protein; CS Na = chondroitin sulfate Sodium; CT = Cartilage thickness; CTV/TV = calcified tissue volume fraction; CTX-II: C-terminal telopeptide of type II collagen; DFO = deferoxamine; D-GlcN = D glucosamine; DH = Dunkin-Hartley; DMMB = dimethylmethylene blue dye; DXA = Dual-energy X-ray absorptiometry; EA = electroacupuncture; EGCG = Epigallocatechin 3-gallate; FI = fibrillation index; Fib3 = Fibulin 3 fragments; FGF18 = fibroblast growth factor 18; FMT = fluorescence molecular tomography; FTH = Ferritin heavy chain; GAG = glycosaminoglycans; H&E = Hematoxylin and Eosin; HA = Hyaluronic Acid; HAMP = hepcidin; HDF = High Fat Diet; hMSCs = human mesenchymal stem cells; HSP70 = heat shock protein 70; IHC = immunohistochemistry; IL = Interleukin; INF = Interferon; IRAP = IL-1 receptor antagonist protein; LIPUS = Low-intensity pulsed ultrasound; MAR = mineral appositional rate; MCP1 = monocyte chemoattractant protein-1; MGP = Matrix Gla (γ-carboxyglutamate) protein; MMA = methylmetacrilate; MMP = metalloproteinase; mo = months; MS/BS = mineralizing surfaces; mTOR = mammalian target of rapamycin; NF-κB p65 = nuclear factor kappa B p65; NLRP3 = nucleotide-binding and oligomerization domain-like receptor containing protein 3; NRF2 = nuclear factor erythroid-2-related factor 2; NTC = non-targeting vector control; NTV = non-targeting control vector; OCT = optimal cutting temperature; OPG = Osteoprotegerin; OS/BS = osteoid-covered surfaces; Pa-MSCs = MSCs isolated from the synovial fluid of OA patients; PBS = phosphate buffered solution; PCNA = proliferating cell nuclear antigen; PEMFs = pulsed electromagnetic fields; PG = Proteoglycans; PGE2 Prostaglandin E2; PRDX1 = peroxiredoxin 1; PRP = Platelet Rich Plasma; PTGS-2 = prostaglandin-endoperoxide synthase 2; PTH (1–34) = parathyroid hormone; PTH1R = PTH receptor; RANTES = regulated on activation, normal T cell expressed and secreted; RANKL = receptor activator of nuclear factor-kB ligand; Rapa = rapamycin; RBC = red blood cells; Re-MSCs = Reprogrammed MSCs; RPS6 = Ribosomal protein S6; SB = Subchondral bone; Sb.St = subchondral bone strength; Sb.Th = subchondral bone thickness; SBT = Subchondral bone thickness; SDF1 = stromal cell derived factor-1; SF = Synovial Fluid; SLC11A2/DMT1 = Divalent metal transporter 1; SLC39A14/ZIP14 = ZRT/IRT-like protein 14; SLC40A1/FPN = Ferroportin; SMI = Structure Model Index; SOD = superoxide dismutase; SOST = sclerostin; SPF = specific pathogen-free; TbSp = Trabecular separation; TbTh = Trabecular Thickness; TFR1 = Transferrin receptor 1; TGFβ1 = Transforming Growth Factor β1; TIMP = tissue inhibitor of metalloproteinases; TNF = Tumor Necrosis Factor; TUNEL = terminal deoxynucleotide transferase-mediated dUTP nick end labeling; TV = targeting knockdown IL1β vector; ULK1 = Unc-51-like kinase 1; WB = Western Blot; WBC = white blood cells; wk = week.

**Table 3 ijms-23-07309-t003:** Risk of bias of in vivo studies.

Ref.	Sequence Generation	Baseline Characteristics	Allocation Concealment	Random Housing	Blinding	Random Outcome Assessment	Blinding	Incomplete Outcome Data	Selective Outcome Reporting	Other Sources of Bias
[12]	NO	YES	NO	NO	NO	YES	YES	YES	YES	UNCLEAR
[13]	NO	YES	NO	NO	NO	NO	NO	YES	YES	UNCLEAR
[15]	NO	YES	NO	NO	NO	NO	NO	YES	YES	UNCLEAR
[17]	NO	YES	NO	NO	NO	NO	NO	YES	YES	UNCLEAR
[18]	NO	YES	NO	NO	NO	NO	YES	YES	YES	UNCLEAR
[19]	NO	YES	NO	NO	NO	NO	NO	YES	YES	UNCLEAR
[20]	NO	YES	NO	NO	NO	YES	NO	YES	YES	UNCLEAR
[22]	NO	YES	NO	NO	NO	NO	YES	YES	YES	UNCLEAR
[24]	NO	YES	NO	NO	YES	NO	YES	YES	YES	UNCLEAR
[25]	NO	YES	NO	NO	NO	NO	NO	YES	YES	UNCLEAR
[26]	NO	YES	NO	NO	NO	YES	YES	YES	YES	UNCLEAR
[27]	NO	YES	NO	NO	NO	NO	YES	YES	YES	UNCLEAR
[28]	YES	YES	YES	YES	YES	YES	YES	YES	YES	YES
[11]	YES	YES	NO	NO	NO	NO	YES	YES	YES	UNCLEAR
[29]	NO	YES	NO	NO	NO	NO	NO	YES	YES	UNCLEAR
[30]	YES	YES	NO	YES	NO	YES	NO	YES	YES	YES
[31]	NO	YES	NO	NO	NO	NO	NO	YES	YES	UNCLEAR
[2]	NO	UNCLEAR	NO	NO	NO	NO	NO	YES	YES	UNCLEAR
[14]	NO	YES	NO	NO	YES	YES	YES	YES	YES	UNCLEAR
[16]	NO	YES	NO	NO	NO	NO	NO	YES	YES	UNCLEAR
[21]	NO	YES	NO	NO	NO	NO	NO	YES	YES	UNCLEAR
[23]	NO	YES	NO	NO	NO	YES	YES	YES	YES	YES
[32]	NO	YES	NO	YES	NO	NO	YES	YES	YES	UNCLEAR
[33]	NO	YES	NO	NO	YES	NO	YES	YES	YES	YES
[34]	YES	YES	YES	YES	NO	YES	NO	YES	YES	UNCLEAR
[35]	NO	YES	NO	NO	NO	NO	YES	YES	YES	UNCLEAR
[36]	NO	YES	NO	YES	NO	NO	YES	YES	YES	UNCLEAR
[37]	YES	YES	YES	YES	NO	YES	YES	YES	YES	UNCLEAR
[38]	NO	YES	NO	NO	NO	NO	YES	YES	YES	UNCLEAR
[39]	NO	YES	NO	NO	NO	NO	NO	YES	YES	UNCLEAR
[40]	NO	YES	NO	NO	NO	YES	NO	YES	YES	UNCLEAR
[41]	YES	YES	YES	YES	NO	YES	NO	YES	YES	UNCLEAR
[42]	NO	YES	NO	YES	NO	NO	YES	YES	YES	UNCLEAR
[43]	NO	YES	YES	YES	NO	NO	YES	YES	YES	UNCLEAR
[44]	NO	YES	YES	YES	NO	NO	YES	YES	YES	UNCLEAR
[45]	YES	YES	YES	YES	NO	YES	YES	YES	YES	UNCLEAR
[46]	NO	YES	NO	YES	YES	YES	YES	YES	YES	UNCLEAR
[47]	NO	YES	NO	YES	NO	NO	YES	YES	YES	UNCLEAR
[48]	NO	YES	NO	YES	NO	NO	YES	YES	YES	UNCLEAR
[49]	NO	YES	NO	NO	NO	YES	NO	YES	YES	UNCLEAR
[50]	NO	YES	NO	YES	NO	NO	YES	YES	YES	UNCLEAR
[51]	NO	YES	YES	YES	NO	YES	YES	YES	YES	UNCLEAR
[52]	NO	YES	NO	NO	NO	NO	NO	YES	YES	UNCLEAR
[53]	NO	YES	NO	YES	NO	NO	NO	YES	YES	UNCLEAR
[54]	NO	YES	NO	YES	NO	NO	NO	YES	YES	UNCLEAR
[55]	NO	YES	NO	YES	NO	YES	NO	YES	YES	UNCLEAR
[56]	NO	YES	NO	YES	NO	YES	NO	YES	YES	UNCLEAR
[57]	NO	YES	NO	YES	NO	NO	YES	YES	YES	UNCLEAR
[58]	NO	YES	NO	YES	NO	NO	YES	YES	YES	UNCLEAR

Sequence generation: Was the allocation sequence adequately generated and applied?; Baseline characteristics: Were the groups similar at baseline or were they adjusted for confounders in the analysis? Allocation concealment: Was the allocation to the different groups adequately concealed during? Random housing: Were the animals randomly housed during the experiment? Blinding: Were the caregivers and/or investigators blinded from knowledge of which intervention each animal received during the experiment? Random outcome assessment: Were animals selected at random for outcome assessment? Blinding: Was the outcome assessor blinded? Incomplete outcome data: Were incomplete outcome data adequately addressed? Selective outcome reporting: Are reports of the study free of selective outcome reporting? Other sources of bias: Was the study apparently free of other problems that could result in high risk of bias? “Yes” indicates low risk of bias; “no” indicates high risk of bias; and “unclear” indicates an unclear risk of bias. If one of the relevant signaling questions is answered with “no,” this indicates high risk of bias for that specific entry.

**Table 4 ijms-23-07309-t004:** Quality assessment of in vivo studies.

Ref.	TITLE	ABSTRACT	INTRODUCTION	METHODS	RESULTS	DISCUSSION	TOTAL
[12]	1	2	3	13	6	5	**30**
[13]	0	2	2	12	2	1	**19**
[15]	1	1	1	11	2	4	**20**
[17]	1	2	3	7	2	2	**17**
[18]	1	2	2	15	4	5	**29**
[19]	1	2	3	10	2	5	**23**
[20]	1	2	3	12	2	3	**23**
[22]	1	2	2	8	2	3	**13**
[24]	1	2	3	12	5	3	**28**
[25]	1	2	3	8	2	5	**21**
[26]	1	2	2	11	2	2	**20**
[27]	1	2	1	11	2	5	**22**
[28]	1	2	3	16	7	5	**34**
[11]	1	2	3	11	6	4	**27**
[29]	1	2	1	8	3	1	**16**
[30]	1	2	3	13	2	2	**23**
[31]	1	2	2	9	1	3	**18**
[2]	1	2	2	9	2	3	**19**
[14]	1	2	3	12	2	1	**22**
[16]	1	2	2	11	2	1	**19**
[21]	1	2	2	11	2	1	**19**
[23]	1	2	3	13	2	5	**26**
[32]	1	2	3	14	2	3	**25**
[33]	1	2	2	10	2	5	**22**
[34]	1	2	3	10	3	1	**20**
[35]	1	2	2	12	1	3	**21**
[36]	1	2	3	12	1	3	**22**
[37]	1	2	3	11	5	4	**26**
[38]	1	2	2	10	1	2	**18**
[39]	1	2	2	9	2	2	**18**
[40]	1	2	3	13	3	4	**26**
[41]	1	2	3	12	2	4	**24**
[42]	1	2	3	13	5	3	**27**
[43]	1	2	3	16	6	4	**32**
[44]	1	2	3	12	7	4	**29**
[45]	0	2	3	16	3	4	**28**
[46]	1	2	3	13	4	3	**26**
[47]	1	2	2	14	5	5	**29**
[48]	1	2	2	12	2	3	**22**
[49]	0	1	2	8	2	1	**14**
[50]	1	2	2	12	2	3	**22**
[51]	1	2	3	16	6	3	**31**
[52]	1	2	2	11	3	3	**22**
[53]	1	2	2	13	3	3	**24**
[54]	0	2	3	13	4	2	**24**
[55]	0	2	2	13	2	2	**21**
[56]	0	2	2	13	2	2	**21**
[57]	0	2	2	12	3	2	**21**
[58]	1	2	3	12	2	1	**21**

## Figures and Tables

**Figure 1 ijms-23-07309-f001:**
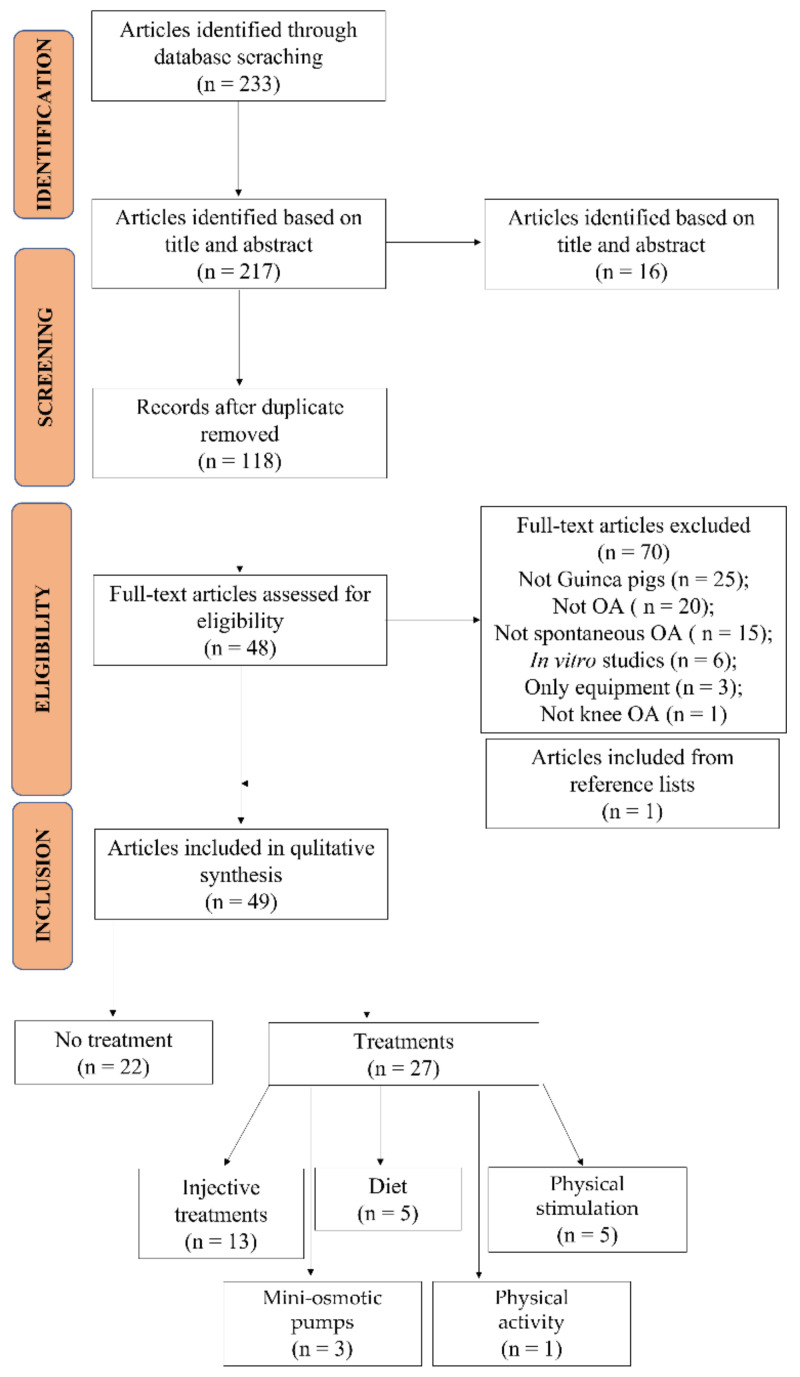
Systematic review flow diagram. The PRISMA flow diagram for the systematic review detailing the database searches, the number of abstracts screened, and the full texts retrieved.

**Figure 2 ijms-23-07309-f002:**
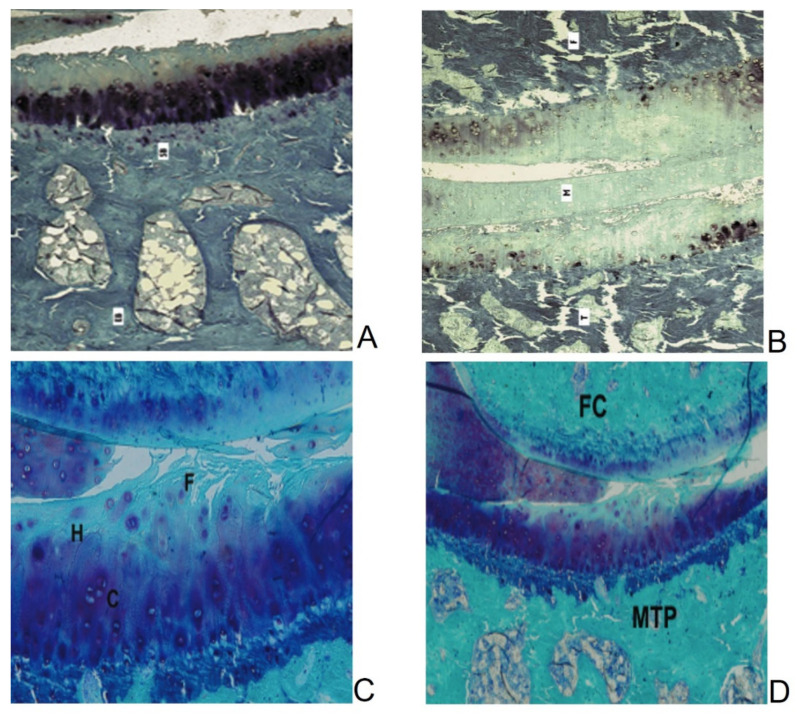
(**A**) Medial tibial plateau and (**B**) medial articular compartment of DH guinea pigs at 12 months of age. Surface irregularities and loss of staining in the superficial and medial cartilage surfaces are visible. (**C**) Medial tibial plateau and (**D**) medial articular compartment of DH guinea pigs at 21 months of age. High fibrillation, reduced staining, and cartilage thickness are observed. SB: subchondral bone, EB: epiphyseal bone, T: medial tibial plateau, F: medial femoral condyle, M: meniscus, H: Hypocellularity zone; C: clusters; FC: femoral condyle, MTP: medial tibial plateau. Toluidine blue/Fast Green staining. (**A**,**C**) 50× of magnification; (**B**,**D**) 8× of magnification.

**Figure 3 ijms-23-07309-f003:**
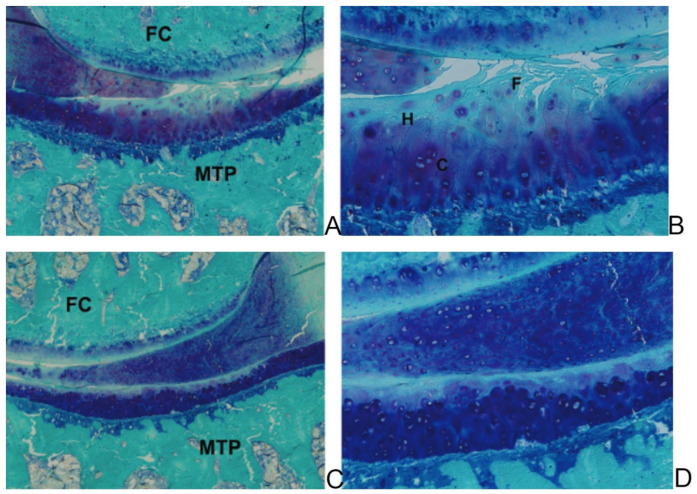
(**A**,**B**) Medial articular compartment of a DH guinea pigs of 21 months of age. High fibrillation, reduced staining and cartilage thickness are observed. (**C**,**D**) Madial articular compartment of a DH guinea pigs of 21 months of age treated with PEMFs at 75 Hz. An increased cartilage preservation is much more evident and reduced fibrillation, increased PG content and normal cell distribution are observed. FC: femoral condyle, MTP: medial tibial plateau, H: Hypocellularity zone, C: clusters, F: fibrillation zone. Toluidine blue/Fast green staining. (**A**,**C**) 8× of magnification; (**B**,**D**) 50× of magnification.

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
