# Peer review of "Naturally Occurring Osteoarthritis Features and Treatments: Systematic Review on the Aged Guinea Pig Model"

_ijms, 2022, doi:10.3390/ijms23137309_

Round 1
Reviewer 1 Report
The manuscript ran out by Veronesi et al. systematically reviewed the features and treatments of primary osteoarthritis (spontaneous osteoarthritis). Interestingly authors have also reviewed the literature and characterized them based on animals age, type of treatments used, and gender-related differences in guinea pig model of OA. Although similar studies have been published to review the animal models of osteoarthritis, this manuscript is an updated version. In addition, the insight of this study through aged guinea pig model of osteoarthritis is novel and may help researchers to find out the suitable model for their projects. In my opinion, this study needs to be improved in terms of various aspects, which I have mentioned them as follows.
1. Since lots of data represented in this manuscript, a graphical abstract may be needed to better understanding the final conclusions/speculates of the authors after reviewing whole literature.
2. Spontaneous OA is a type of primary OA model and it can subcategorize to genetically and naturally occurring. Since this study did not provide any details about this issue, it is better to mention this issue in title, abstract, introduction (aim of the research), and methods. For example, Yan et al. (reference No. 21) have evaluated “Naturally Occurring Osteoarthritis”. So, this issue should be discussed more in details in this manuscript and authors should mention that they focused on naturally occurring of age-related spontaneous OA or following genetic modifications. In my opinion, a comparison between naturally occurring and genetic modified model will be really interesting.
3. Lots of unnecessary information has been provided in Introduction section. So, I recommend author to shrink the introduction and eliminate the parts unrelated to aging and OA. The pathophysiological pathway should be summarized in introduction and expand in discussion. In my opinion, authors should more specifically focused on the aim of this study “age-related spontaneous OA” and cons and pros of different animal models.
4. Authors have mentioned that they excluded studies if they were not related to knee OA. Guinea pigs typically develop knee OA and almost all of researches focused on the knee OA. So, in my opinion, it is better to list all studies have been excluded in each section as a supplementary file. This list would help future studies to easily focus on other elements.
5. Authors have evaluated the risk of bias, however, no strict information and details about scoring have been provided.
6. Figure 1 shows the flow chart of study election. At the bottom of this flow chart, it is better to divide the included studies based on subgroups. For example, please mention the total number of studies used in each section (age, diet, treatment etc.).
7. Authors should provide a new table based on the aim of the manuscript and the correspondence results. For example, categorized the studies that evaluate an imaging method or evaluate the efficacy of specific treatment or a pathophysiological pathway. Although, some of this information may be provided in Table 1, lots of details have been mention in this table and may result in misunderstanding. In other words, this study needs a straight forward table that can categorize the studies involved in this manuscript based on the field of that study. This would be very helpful for readers to understand the role of this animal model in a specific field (e.g. imaging or pharmacology).
8. Each section of results needs a table/graph for better understanding. Although tables 1, 2, and 3 can provide valuable information, they can use as supplementary. In my opinion, designing new tables and also used previously published figures of OA in Guinea Pig (authors can use the published figures by getting a permission from the publisher) in each section of “Results” can robustly improve this manuscript.
Author Response
The manuscript ran out by Veronesi et al. systematically reviewed the features and treatments of primary osteoarthritis (spontaneous osteoarthritis). Interestingly authors have also reviewed the literature and characterized them based on animals age, type of treatments used, and gender-related differences in guinea pig model of OA. Although similar studies have been published to review the animal models of osteoarthritis, this manuscript is an updated version. In addition, the insight of this study through aged guinea pig model of osteoarthritis is novel and may help researchers to find out the suitable model for their projects. In my opinion, this study needs to be improved in terms of various aspects, which I have mentioned them as follows.
- Since lots of data represented in this manuscript, a graphical abstract may be needed to better understanding the final conclusions/speculates of the authors after reviewing whole literature.
As reviewer suggested, now a graphical abstract has been added.
- Spontaneous OA is a type of primary OA model and it can subcategorize to genetically and naturally occurring. Since this study did not provide any details about this issue, it is better to mention this issue in title, abstract, introduction (aim of the research), and methods. For example, Yan et al. (reference No. 21) have evaluated “Naturally Occurring Osteoarthritis”. So, this issue should be discussed more in details in this manuscript and authors should mention that they focused on naturally occurring of age-related spontaneous OA or following genetic modifications. In my opinion, a comparison between naturally occurring and genetic modified model will be really interesting.
As was rightly said by the reviewer, in animal models, OA is divided into 2 categories: 1) the spontaneous OA, in turn divided into naturally occurring (primary OA) and generated by genetic abnormalities; 2) the induced OA, through a surgical procedure (secondary OA) or intra-articular injection of chemical products.
In literature several different animal models are used for the in vivo evaluations of OA, and they are small and large models. Small animals (mice, rats, guinea pigs, and rabbits) are employed for the evaluation of pathogenesis and pathophysiology of OA, since they show more rapid disease progression, and are relatively inexpensive and easy to handle.
Large animals (dog, goat/sheep, and horse) are mostly used to study the OA process and treatment. Due to their anatomical similarity with humans, these animals confirm the efficacy of a treatment before clinical employment in trials.
As regard small animal models, they are differently engaged to evaluate spontaneous or induced OA. Among them, mice are employed to better understand the genetic bases of OA, rats are more considered for the evaluation of chondrocyte metabolism, due to metabolic abnormalities, and rabbits for tissue bioengineering and focal osteochondral lesions studies.
In addition, rabbits, rats, and mice are also useful in studies of induced OA models, through anterior cruciate ligament transection (ACLT), meniscal destabilization, collateral ligament transection, and of chemical induction models with monosodium iodoacetate (MIA), papain, collagenase, and quinolones.
The spontaneous OA models show a slow evolution, leading to high lengthy and economic costs, but they have a good correlation with the natural progression of human OA. Among them the naturally occurring ones are used to evaluate the pathogenesis of OA, while the genetic models are the result of alterations in genes involved in degradation of the cartilage matrix, in chondrocyte differentiation or apoptosis, in subchondral bone metabolism, and in
inflammatory molecules. The genetic models accelerate OA development, but they could induce inter-animal variability and are less representative of the condition present in humans in which several different genetic pathways are modified.
Guinea pigs are used as naturally developed models of OA, in which the effect of age, diet and sex are investigated. Guinea pigs are the most useful for the evaluation of human primary OA and of inflammatory biomarkers for their histopathological and OA lesion similarities with humans. In addition, they quickly reach maturity.
Based on the reviewer's suggestion, title, abstract, introduction (aim of the research), and methods have been modified as follows: “Naturally Occurring Osteoarthritis Features And Treatments: Systematic Review On The Aged Guinea Pig Model” (Title);
“To date, several in vivo models are used to reproduce the onset and monitor the progression of osteoarthritis (OA) and guinea pigs represent a standard model for study naturally occurring, age-related OA. This systematic review aims to characterize guinea pig for its employment in in vivo naturally occurring OA studies and for the evaluation of specific disease-modifying agents” (Abstract);
“Spontaneous OA due to aging, represent the most common form of naturally occurring OA in humans [2, 3].
The 2 categories of OA also in in vivo preclinical models are: 1) the spontaneous OA, in turn divided into naturally occurring (primary OA) and generated by genetic abnormalities; 2) the induced OA, through a surgical procedure (secondary OA) or intra-articular injection of chemical products [7, 8].
Small and large animal models are used. Small animals (mice, rats, guinea pigs, and rabbits) are employed for the evaluation of pathogenesis and pathophysiology of OA, since they show more rapid disease progression, and are relatively inexpensive and easy to handle.
Large animals (dog, goat/sheep, and horse) are mostly used to study the OA process and treatment. Due to their anatomical similarity with humans, these animals confirm the efficacy of a treatment before clinical employment in trials.
As regard small animal models, they are differently engaged to evaluate spontaneous or induced OA. Among them, mice are employed to better understand the genetic bases of OA, rats are more considered for the evaluation of chondrocyte metabolism, due to metabolic abnormalities, and rabbits for tissue bioengineering and focal osteochondral lesions studies.
In addition, rabbits, rats, and mice are also useful in studies of induced OA models, through anterior cruciate ligament transection (ACLT), meniscal destabilization, collateral ligament transection, and of chemical induction models with monosodium iodo-acetate (MIA), papain, collagenase, and quinolones.
The spontaneous OA models show a slow evolution, leading to high lengthy and eco-nomic costs, but they have a good correlation with the natural progression of human OA. The naturally occurring ones are used to evaluate the pathogenesis of OA, while the genetic models are the result of alterations in genes involved in degradation of the cartilage matrix, in chondrocyte differentiation or apoptosis, in subchondral bone metabolism, and in inflammatory molecules. The genetic models accelerate OA development, but they could induce inter-animal variability and are less representative of the condition present in humans in which several different genetic pathways are modified [3, 7, 8].
Therefore, the aim of the present review focused on naturally occurring OA and we systematically characterized guinea pig animal model for its employment in in vivo studies.” (Introduction);
“….(2) studies that evaluate spontaneous naturally occurring OA (interventions)….” (Materials and Methods).
In addition, new references have been added in the “References” paragraph as follows: “3. Serra, C.I., Soler, C. Animal Models of Osteoarthritis in Small Mammals. Vet Clin North Am Exot Anim Pract. 2019;22(2):211-221.
- Kuyinu, E.L., Narayanan, G., Nair, L.S., Laurencin, C.T. Animal models of osteoarthritis: classification, update, and meas-urement of outcomes. J Orthop Surg Res. 2016;11:19.
- Bapat, S., Hubbard, D., Munjal, A., Hunter, M., Fulzele, S. Pros and cons of mouse models for studying osteoarthritis. Clin Transl Med. 2018;7:36.” (page 26, lines 377-378; page 27, lines 387-390).
- Lots of unnecessary information has been provided in Introduction section. So, I recommend author to shrink the introduction and eliminate the parts unrelated to aging and OA. The pathophysiological pathway should be summarized in introduction and expand in discussion. In my opinion, authors should more specifically focused on the aim of this study “age-related spontaneous OA” and cons and pros of different animal models.
We thank the reviewer for this advice, so we modified the “Introduction” paragraph and some sentences, deleted from the “Introduction” paragraph, have been reframed in the “Discussion” paragraph as follows: “More than 250 million of individuals are affected worldwide and nearly 10-15% of people over 60 years of age suffer from OA with pain and disability. Even if OA can affect all joints, hip and knee are the most affected ones (⁓3.8% of the population worldwide) [1, 64].
Although there is no real knowledge about the order of events leading up to OA, the SB thickening seems one of the first event involved. SB mineral density is higher in OA patients and leads to sclerosis and osteophyte formation, cartilage degeneration and defects, joint space narrowing and altered structure and biochemical properties of menisci [65]. Subsequently, hypertrophy and apoptosis of chondrocytes happens, with a resulting degradation in cartilage matrix, particularly in Type II Collagen (COLL II). Additionally, extracellular matrix (ECM) degrading enzymes and inflammatory cytokines, such as matrix metalloproteinases (MMPs), a disintegrin and metalloproteinase with thrombospondin type 1 motif 5 (ADAMTS5), interleukin-1 (IL-1), necrosis factor alpha (TNF-α), and nuclear factor kap-pa-light-chain-enhancer of activated B cells (NF-κB) increased [66, 67]. NF-κB acts as a transcription factor that induces gene transcription of many inflammatory cytokines. When it translocates into the nucleus, it triggers the consequent transcription of inflammatory mediators and catabolic enzymes, such as interleukin (IL)-6, IL-8, and MMPs. The pro-longed inflammation provokes the loss of the growth-arrested state of articular chondrocytes, deregulated gene expression, and consequent cartilage degradation of OA [68, 69].” (Page 24, lines 226-245).
In addition, new references have been added to the whole manuscript and in the “References” paragraph: “68. Veronesi, F., Giavaresi, G., Maglio, M., Scotto d'Abusco, A., Politi, L., Scandurra, R., Olivotto, E., Grigolo, B., Borzì, R.M., Fini, M. Chondroprotective activity of N-acetyl phenylalanine glucosamine derivative on knee joint structure and in-flammation in a murine model of osteoarthritis. Osteoarthritis Cartilage. 2017;25(4):589-599.
Vassallo, V., Stellavato, A., Cimini, D., Pirozzi, A.V.A., Alfano, A., Cammarota, M., Balato, G., D'Addona, A., Ruosi, C., Schiraldi, C. Unsulfated biotechnological chondroitin by itself as well as in combination with high molecular weight hya-luronan improves the inflammation profile in osteoarthritis in vitro model. J Cell Biochem. 2021;122(9):1021-1036.” (page 29, lines 529-534).
4. Authors have mentioned that they excluded studies if they were not related to knee OA. Guinea pigs typically develop knee OA and almost all of researches focused on the knee OA. So, in my opinion, it is better to list all studies have been excluded in each section as a supplementary file. This list would help future studies to easily focus on other elements.
As observed in Figure 1, some studies (n = 70) were excluded from the list of accepted included studies because they:
- were performed in animals other than guinea pigs, such as rats (n = 7), rabbits (n = 6), mice (n = 5), bovines (n = 2), humans (n = 2), mini-pigs (n = 1), sheep (n = 1), and pigs (n = 1);
- did not treat the OA pathology, but skeletal muscle injury (n = 3), costal cartilage injury (n = 2), mitochondrial genome sequencing (n = 1), plasma and joint tissue pharmacokinetics (n = 1), arterial hypertension and hyperlipidemia (n = 1), intervertebral disc degeneration (n = 1), hematology (n = 1), hemorrhagic fever (n = 1), muscle afferent neurones disease (n = 1), osteochondral defects (n = 2), ex vivo studies in humans and bovine cartilage (n = 3), finite element (n = 3);
- treated non-spontaneous OA, but induced through systemic iron overload (n = 1), meniscectomy (n = 7), cranial cruciate ligament excision (n = 3), and anterior cruciate ligament transaction (ACLT) (n = 4);
- were in vitro studies on chondrocytes (n = 2), synoviocytes (n = 1), immune cells (n = 1), and meniscal cells (n = 2);
- described equipments for diagnostic purposes, such as a high-resolution small animal ultrasound system with a transducer for image-guided injections (n = 1), and MRI for animal research (n = 2);
- did not treat knee OA, but temporomandibular one (n = 1)
Now in a supplementary file the following sentences have been added:
“As observed in Figure 1, some studies (n = 70) were excluded from the list of accepted included studies because they:
- were performed in animals other than guinea pigs, such as rats (n = 7), rabbits (n = 6), mice (n = 5), bovines (n = 2), humans (n = 2), mini-pigs (n = 1), sheep (n = 1), and pigs (n = 1);
- did not treat the pathology of OA, but skeletal muscle injury (n = 3), costal cartilage injury (n = 2), mitochondrial genome sequencing (n = 1), plasma and joint tissue pharmacokinetics (n = 1), arterial hypertension and hyperlipidemia (n = 1), intervertebral disc degeneration (n = 1), hematology (n = 1), hemorrhagic fever (n = 1), muscle afferent neurones disease (n = 1), osteochondral defects (n = 2), ex vivo studies in humans and bovine cartilage (n = 3), finite element (n = 3);
- treated non-spontaneous OA, but induced through systemic iron overload (n = 1), meniscectomy (n = 7), cranial cruciate ligament excision (n = 3), and anterior cruciate ligament transaction (ACLT) (n = 4);
- were in vitro studies on chondrocytes (n = 2), synoviocytes (n = 1), immune cells (n = 1), and meniscal cells (n = 2);
- described equipments for diagnostic purposes, such as a high-resolution small animal ultrasound system with a transducer for image-guided injections (n = 1), and MRI for animal research (n = 2);
- did not treat knee OA, but temporomandibular one (n = 1)” (Supplementary file).
- Authors have evaluated the risk of bias, however, no strict information and details about scoring have been provided.
Accordingly to the reviewer’s suggestion, information about the score employed to evaluate risk of bias have been added to the current Table 3 as followed: “Sequence generation: Was the allocation sequence adequately generated and applied?; Baseline characteristics: Were the groups similar at baseline or were they adjusted for confounders in the analysis?; Allocation concealment: Was the allocation to the different groups adequately concealed during?; Random housing: Were the animals randomly housed during the experiment?; Blinding: Were the caregivers and/or investigators blinded from knowledge which intervention each animal received during the experiment?; Random outcome assessment: Were animals selected at random for outcome assessment?; Blinding: Was the outcome assessor blinded?; Incomplete outcome data: Were incomplete outcome data adequately addressed?; Selective outcome reporting: Are reports of the study free of selective outcome reporting?; Other sources of bias: Was the study apparently free of other problems that could result in high risk of bias?.
“Yes” indicates low risk of bias; “no” indicates high risk of bias; and “unclear” indicates an unclear risk of bias. If one of the relevant signaling questions is answered with “no,” this indi-cates high risk of bias for that specific entry.”
It was detailed in Reference number 15 [15.Hooijmans, C.R., Rovers, M.M., de Vries, R.B.M., Leenaars, M., Ritskes-Hoitinga, M., Langendam, M.W. SYRCLE’s risk of bias tool for animal studies. BMC Med Res Methodol. 2014;14:43].
In “Materials and methods” paragraph the following sentence has been added: “The results of risk of Bias [15] and Quality assessment [16] of the in vivo studies were summarized in Tables 3 and 4.” (page 3, lines 130-131).
- Figure 1 shows the flow chart of study election. At the bottom of this flow chart, it is better to divide the included studies based on subgroups. For example, please mention the total number of studies used in each section (age, diet, treatment etc.).
Accordingly to the reviewer’s suggestion, Figure 1 has been modified.
- Authors should provide a new table based on the aim of the manuscript and the correspondence results. For example, categorized the studies that evaluate an imaging method or evaluate the efficacy of specific treatment or a pathophysiological pathway. Although, some of this information may be provided in Table 1, lots of details have been mention in this table and may result in misunderstanding. In other words, this study needs a straight forward table that can categorize the studies involved in this manuscript based on the field of that study. This would be very helpful for readers to understand the role of this animal model in a specific field (e.g. imaging or pharmacology).
As suggested by the reviewer, Table 1 has been substantially modified. First of all it has been divided into 2 tables, now Table 1 and Table 2, to group the studies that evaluated “Preclinical in vivo studies evaluating naturally occurring OA in guinea pigs.” (Table 1) and “Preclinical in vivo studies evaluating treatments in naturally occurring OA in DH guinea pigs.” (Table 2).
The Tables have also been simplified compared to the original table by removing all excess information and the aim of each study has been added. In addition, as observed in Table 1, the studies have been further grouped in those that evaluated the changes of joint structure during naturally occurring OA and those that also evaluated specific pathways, such as TGFβ, PPARγ, and H- and L-PGDS, and IGF-1 ones, and the levels of cartilage MGP and serum estradiol. The studies of the Table 2 are further divided into those that evaluated injective treatments, diet, physical stimulation, mini-osmotic pumps, and physical activity.
In the whole manuscript the Tables have been renumbered.
- Each section of results needs a table/graph for better understanding. Although tables 1, 2, and 3 can provide valuable information, they can use as supplementary. In my opinion, designing new tables and also used previously published figures of OA in Guinea Pig (authors can use the published figures by getting a permission from the publisher) in each section of “Results” can robustly improve this manuscript.
The tables have been modified accordingly to the previous reviewer’s suggestion and in particular, Table 1 and Table 2 have been changed also considering the results present in each results section. In addition, now new histological images of guinea pigs have been added for each results section for better understanding. We decided to include explanatory figures that show the changes in the knee joint during DH guinea pigs aging (Figure 2) and the efficacy of one of the treatments, such as PEMF stimulation (Figure 3).
The Figure 2 is added to the manuscript: “Figure 2 shows knee joints of DH guinea pigs at two different months of age (12 and 21 months), demonstrating different OA stages.” (page 18, lines 19-20) and the figure caption is the followed: “Figure 2. A) Medial tibial plateau and B) Medial articular compartment of a DH guinea pigs of 12 months of age. Surface irregularities and loss of staining in the superficial and medial cartilage surfaces are visible. C) Medial tibial plateau and D) Medial articular compartment of a DH guinea pigs at 21 months of age. High fibrillation, reduced staining and cartilage thickness are observed. SB: subchondral bone, EB: epiphyseal bone, T: medial tibial plateau, F: medial femoral condyle, M: meniscus, H: Hypocellularity zone; C: clusters; FC: femoral condyle, MTP: medial tibial plateau. Toluidine blue/Fast Green staining. A), C) 50X of magnifi-cation; B), D) 8X of magnification.” (page 20, lines 80-86).
The Figure 3 is added to the manuscript: “Figure 3 shows knee joints of DH guinea pigs at 21 months treated or not with PEMF stimulation.” (page 22, lines 173-174) and the Figure caption is the followed: “Figure 3. A), B) Medial articular compartment of a DH guinea pigs of 21 months of age. High fibrillation, reduced staining and cartilage thickness are observed.
C), D) Madial articular compartment of a DH guinea pigs of 21 months of age treated with PEMFs at 75 Hz. An increased cartilage preservation is much more evident and reduced fibrillation, increased PG content and normal cell distribution are observed.
FC: femoral condyle, MTP: medial tibial plateau, H: Hypocellularity zone, C: clusters, F: fibrillation zone. Toluidine blue/Fast green staining. A), C) 8X of magnification; B), D) 50X of magnification” (page 23, lines 203-209).
Reviewer 2 Report
In the review :“Spontaneous Osteoarthritis Features And Treatments: Systematic Review On The Aged Guinea Pig Mod” the authors explained the potential employment of guinea pig for spontaneous OA studies and for the evaluation of specific disease-modifying agents.
The authors clearly explain the rational of the study and the different field of application of guinea pigs. However, we would like to invite the authors to clarify some minor points:
1. In the introduction the authors explain the role of some biomarkers such as Type II Collagen (COLL II), metalloproteinases (MMPs), a disintegrin and metalloproteinase with thrombospondin type 1 motif 5 (ADAMTS5), interleukin-1 (IL-1), and necrosis factor alpha (TNF-α). However, they should also introduce the role of NF-kB into OA disease.
2. Among the introduction of OA studies through in vitro model it should be inserted a reference. The following could be useful: Vassallo V, Stellavato A, Cimini D, Pirozzi AVA, Alfano A, Cammarota M, Balato G, D'Addona A, Ruosi C, Schiraldi C. Unsulfated biotechnological chondroitin by itself as well as in combination with high molecular weight hyaluronan improves the inflammation profile in osteoarthritis in vitro model. J Cell Biochem. 2021 May 31;122(9):1021–36. doi: 10.1002/jcb.29907. Epub ahead of print. PMID: 34056757; PMCID: PMC8453819.
3. Please insert more detail about the use of this type of animal instead of other (mice, rats, rabbits).
4
Author Response
In the review: “Spontaneous Osteoarthritis Features And Treatments: Systematic Review On The Aged Guinea Pig Mod” the authors explained the potential employment of guinea pig for spontaneous OA studies and for the evaluation of specific disease-modifying agents.
The authors clearly explain the rational of the study and the different field of application of guinea pigs. However, we would like to invite the authors to clarify some minor points:
- In the introduction the authors explain the role of some biomarkers such as Type II Collagen (COLL II), metalloproteinases (MMPs), a disintegrin and metalloproteinase with thrombospondin type 1 motif 5 (ADAMTS5), interleukin-1 (IL-1), and necrosis factor alpha (TNF-α). However, they should also introduce the role of NF-kB into OA disease.
As rightly said by the reviewer, among the several different biomarkers of OA, also NF-kB plays a pivotal role in OA pathology.
In our two previous published preclinical studies, we evaluated NF-kB after a glucosamine (GlcN) derivative treatment in mice model of OA and in chondrocyte-synoviocyte cocultures that mimicked an OA microenvironment in vitro [PMID: 31537813 and PMID: 27836674].
Nuclear factor kappa-light-chain-enhancer of activated B cells (NF-κB) acts as a transcription factor, inducing gene transcription of many inflammatory cytokines. The removal of IκB inhibitors from the cytoplasmic NF-κB complex by some kinases, allows the translocation of NF-κB into the nucleus triggering the consequent transcription of inflammatory mediators and catabolic enzymes, such as interleukin (IL)-6, IL-8, and metalloproteinases (MMPs). The prolonged inflammation provokes the loss of the growth-arrested state of articular chondrocytes, deregulated gene expression, and consequent cartilage degradation.
So, also based on the suggestions of the other reviewer, the part of biomarkers presents in the “Introduction” paragraph has been transferred to the “Discussion” paragraph, and new sentences have been added as follows: “Additionally, extracellular matrix (ECM) degrading enzymes and inflammatory cytokines, such as matrix metalloproteinases (MMPs), a disintegrin and metalloproteinase with thrombospondin type 1 motif 5 (ADAMTS5), interleukin-1 (IL-1), necrosis factor alpha (TNF-α), and nuclear factor kappa-light-chain-enhancer of activated B cells (NF-κB) in-creased [66, 67]. NF-κB acts as a transcription factor that induces gene transcription of many inflammatory cytokines. When it translocates into the nucleus, it triggers the conse-quent transcription of inflammatory mediators and catabolic enzymes, such as interleukin (IL)-6, IL-8, and MMPs. The prolonged inflammation provokes the loss of the growth-arrested state of articular chondrocytes, deregulated gene expression, and conse-quent cartilage degradation of OA [68, 69].” (page 24, lines 235-245).
In addition, 2 new references have been added. “68. Veronesi, F., Giavaresi, G., Maglio, M., Scotto d'Abusco, A., Politi, L., Scandurra, R., Olivotto, E., Grigolo, B., Borzì, R.M., Fini, M. Chondroprotective activity of N-acetyl phenylalanine glucosamine derivative on knee joint structure and inflammation in a murine model of osteoarthritis. Osteoarthritis Cartilage. 2017;25(4):589-599.
- Vassallo, V., Stellavato, A., Cimini, D., Pirozzi, A.V.A., Alfano, A., Cammarota, M., Balato, G., D'Addona, A., Ruosi, C., Schiraldi, C. Unsulfated biotechnological chondroitin by itself as well as in combination with high molecular weight hyaluronan improves the inflammation profile in osteoarthritis in vitro model. J Cell Biochem. 2021;122(9):1021-1036.” (page 29, lines 529-534).
2. Among the introduction of OA studies through in vitro model it should be inserted a reference. The following could be useful: Vassallo V, Stellavato A, Cimini D, Pirozzi AVA, Alfano A, Cammarota M, Balato G, D'Addona A, Ruosi C, Schiraldi C. Unsulfated biotechnological chondroitin by itself as well as in combination with high molecular weight hyaluronan improves the inflammation profile in osteoarthritis in vitro model. J Cell Biochem. 2021 May 31;122(9):1021–36. doi: 10.1002/jcb.29907. Epub ahead of print. PMID: 34056757; PMCID: PMC8453819.
This paper has now been added in the main text as follows: “Several experimental studies highlighted the capabilities of in vitro 3D culture systems to recapitulate the characteristic of a pathology [71, 69].” (page 24, line 255) and in the “references” paragraph as follows: “69. Vassallo, V., Stellavato, A., Cimini, D., Pirozzi, A.V.A., Alfano, A., Cammarota, M., Balato, G., D'Addona, A., Ruosi, C., Schiraldi, C. Unsulfated biotechnological chondroitin by itself as well as in combination with high molecular weight hyaluronan improves the inflammation profile in osteoarthritis in vitro model. J Cell Biochem. 2021;122(9):1021-1036.” (page 29, lines 532-534).
Section about in vitro models has been transferred from the “Introduction” paragraph to “Discussion” one, at the request of the reviewer 1.
- Please insert more detail about the use of this type of animal instead of other (mice, rats, rabbits).
OA is divided into 2 categories: 1) the spontaneous OA, in turn divided into naturally occurring (primary OA) and generated by genetic abnormalities; 2) the induced OA, through a surgical procedure (secondary OA) or intra-articular injection of chemical products.
In literature several different animal models are used for the in vivo evaluations of OA, and they are small and large models. Small animals (mice, rats, guinea pigs, and rabbits) are employed for the evaluation of pathogenesis and pathophysiology of OA, since they show more rapid disease progression, and are relatively inexpensive and easy to handle.
Large animals (dog, goat/sheep, and horse) are mostly used to study the OA process and treatment. Due to their anatomical similarity with humans, these animals confirm the efficacy of a treatment before clinical employment in trials.
As regard small animal models, they are differently engaged to evaluate spontaneous or induced OA. Among them, mice are employed to better understand the genetic bases of OA, rats are more considered for the evaluation of chondrocyte metabolism, due to metabolic abnormalities, and rabbits for tissue bioengineering and focal osteochondral lesions studies.
In addition, rabbits, rats, and mice are also useful in studies of induced OA models, through anterior cruciate ligament transection (ACLT), meniscal destabilization, collateral ligament transection, and of chemical induction models with monosodium iodoacetate (MIA), papain, collagenase, and quinolones.
The spontaneous OA models show a slow evolution, leading to high lengthy and economic costs, but they have a good correlation with the natural progression of human OA. Among them the naturally occurring ones are used to evaluate the pathogenesis of OA, while the genetic models are the result of alterations in genes involved in degradation of the cartilage matrix, in chondrocyte differentiation or apoptosis, in subchondral bone metabolism, and in inflammatory molecules. The genetic models accelerate OA development, but they could induce inter-animal variability and are less representative of the condition present in humans in which several different genetic pathways are modified.
Guinea pigs are used as naturally developed models of OA, in which the effect of age, diet and sex are investigated. Guinea pigs are the most useful for the evaluation of human primary OA and of inflammatory biomarkers for their histopathological and OA lesion similarities with humans. In addition, they quickly reach maturity.
So, in the “Introduction” paragraph, the following sentences have been added as follows: “The 2 categories of OA also in in vivo preclinical models are: 1) the spontaneous OA, in turn divided into naturally occurring (primary OA) and generated by genetic ab-normalities; 2) the induced OA, through a surgical procedure (secondary OA) or in-tra-articular injection of chemical products [7, 8].
Small and large animal models are used. Small animals (mice, rats, guinea pigs, and rabbits) are employed for the evaluation of pathogenesis and pathophysiology of OA, since they show more rapid disease progression, and are relatively inexpensive and easy to handle.
Large animals (dog, goat/sheep, and horse) are mostly used to study the OA process and treatment. Due to their anatomical similarity with humans, these animals confirm the efficacy of a treatment before clinical employment in trials.
As regard small animal models, they are differently engaged to evaluate spontaneous or induced OA. Among them, mice are employed to better understand the genetic bases of OA, rats are more considered for the evaluation of chondrocyte metabolism, due to metabolic abnormalities, and rabbits for tissue bioengineering and focal osteochondral lesions studies.
In addition, rabbits, rats, and mice are also useful in studies of induced OA models, through anterior cruciate ligament transection (ACLT), meniscal destabilization, collat-eral ligament transection, and of chemical induction models with monosodium iodo-acetate (MIA), papain, collagenase, and quinolones.
The spontaneous OA models show a slow evolution, leading to high lengthy and eco-nomic costs, but they have a good correlation with the natural progression of human OA. The naturally occurring ones are used to evaluate the pathogenesis of OA, while the genetic models are the result of alterations in genes involved in degradation of the car-tilage matrix, in chondrocyte differentiation or apoptosis, in subchondral bone metabo-lism, and in inflammatory molecules. The genetic models accelerate OA development, but they could induce inter-animal variability and are less representative of the condi-tion present in humans in which several different genetic pathways are modified [3, 7, 8].” (page 2., lines 46-74).
New references have been added as follows: “3.Serra, C.I., Soler, C. Animal Models of Osteoarthritis in Small Mammals. Vet Clin North Am Exot Anim Pract. 2019;22(2):211-221.
- Kuyinu, E.L., Narayanan, G., Nair, L.S., Laurencin, C.T. Animal models of osteoarthritis: classification, update, and meas-urement of outcomes. J Orthop Surg Res. 2016;11:19.
- Bapat, S., Hubbard, D., Munjal, A., Hunter, M., Fulzele, S. Pros and cons of mouse models for studying osteoarthritis. Clin Transl Med. 2018;7:36.” (page 26, lines 377-378; page 27, lines 387-390).
Round 2
Reviewer 1 Report
The revision made by authors is satisfactory and authors have addressed all of my comments. However, I did not find out the graphical abstract, please provide the graphical abstract prior to publication.